# Dynamic 1D search and processive nucleosome translocations by RSC and ISW2 chromatin remodelers

Jee Min Kim[1†], Claudia C Carcamo[2†], Sina Jazani[2], Zepei Xie[1], Xinyu A Feng[1,2], Maryam Yamadi[1], Matthew Poyton[1], Katie L Holland[3], Jonathan B Grimm[3], Luke D Lavis[3], Taekjip Ha[2,4*], Carl Wu[1,5*]

[1]Department of Biology, Johns Hopkins University, Baltimore, United States; [2]Department of Biophysics and Biophysical Chemistry, Johns Hopkins University School of Medicine, Baltimore, United States; [3]Janelia Research Campus, Howard Hughes Medical Institute, Ashburn, United States; [4]Howard Hughes Medical Institute, Boston, United States; [5]Department of Molecular Biology and Genetics, Johns Hopkins School of Medicine, Baltimore, United States

**\*For correspondence:**
taekjip.ha@childrens.harvard.
edu (TH);
wuc@jhu.edu (CW)

[†]These authors contributed equally to this work

**Abstract** Eukaryotic gene expression is linked to chromatin structure and nucleosome positioning by ATP-dependent chromatin remodelers that establish and maintain nucleosome-depleted regions (NDRs) near transcription start sites. Conserved yeast RSC and ISW2 remodelers exert antagonistic effects on nucleosomes flanking NDRs, but the temporal dynamics of remodeler search, engagement, and directional nucleosome mobilization for promoter accessibility are unknown. Using optical tweezers and two-color single-particle imaging, we investigated the Brownian diffusion of RSC and ISW2 on free DNA and sparse nucleosome arrays. RSC and ISW2 rapidly scan DNA by one-dimensional hopping and sliding, respectively, with dynamic collisions between remodelers followed by recoil or apparent co-diffusion. Static nucleosomes block remodeler diffusion resulting in remodeler recoil or sequestration. Remarkably, both RSC and ISW2 use ATP hydrolysis to translocate mono-nucleosomes processively at ~30 bp/s on extended linear DNA under tension. Processivity and opposing push–pull directionalities of nucleosome translocation shown by RSC and ISW2 shape the distinctive landscape of promoter chromatin.

## eLife assessment

This manuscript describes **fundamental** single-molecule correlative force and fluorescence microscopy experiments to visualize the 1D diffusion dynamics and long-range nucleosome sliding activity of the yeast chromatin remodelers, RSC and ISW2. **Compelling** evidence shows that both remodelers exhibit 1D diffusion on bare DNA but utilize different mechanisms, with RSC primarily hopping and ISW2 mainly sliding on DNA. These results will be of interest to researchers working on chromatin remodeling.

## Introduction

Eukaryotic chromatin structure is central to gene expression, with nucleosome positioning and composition being established and maintained by four families of ATP-dependent chromatin remodelers (SWI/SNF, CHD, ISWI, and INO80) (*Becker and Workman, 2013*; *Rando and Winston, 2012*; *Zhang et al., 2011*). Active or poised gene promoters exhibit a defined chromatin architecture, the nucleosome-free or nucleosome-depleted region (NDR) (*Yuan et al., 2005*; *Albert et al., 2007*) to

which transcriptional machinery is recruited (*Rando and Winston, 2012*). Genomic studies have shown that in budding yeast, RSC (SWI/SNF family) and ISW2 (ISWI family) remodelers have opposing directional effects on promoter nucleosome movements that widen or narrow the NDR with corresponding effects on transcription in vivo, giving rise to the concept of dynamic nucleosome pushing and pulling by remodelers to regulate promoter accessibility (*Yen et al., 2012*; *Kubik et al., 2018*; *Prajapati et al., 2020*; *Ng et al., 2002*). Remodelers essential for repositioning promoter nucleosomes and incorporating histone variant H2A.Z are enriched in vivo at the +1 and −1 nucleosomes and the intervening NDR DNA (; *Kubik et al., 2019*).

Live-cell single-particle tracking (SPT) studies have provided a window into the complexity of remodeler–chromatin interactions, revealing substantial binding frequencies, highly transient chromatin association, and stable residence times of only several seconds (*Lionnet and Wu, 2021*; *Kim et al., 2021*). By integrating kinetic SPT findings with genomic and proteomic data, remodelers with opposing functions could be calculated to co-occupy the same promoter DNA at some frequency, suggesting a 'tug-of-war' competition between remodelers for the +1 and −1 nucleosomes flanking the NDR (*Kim et al., 2021*). Moreover, chromatin-bound remodelers display several-fold higher diffusion coefficients than nucleosomal histones in living cells, indicating potential one-dimensional (1D) diffusion of bound-state remodelers, i.e., local 1D search on nucleosome-depleted promoter chromatin (*Kim et al., 2021*) similar to 1D diffusion of bacterial DNA-binding proteins (*Berg et al., 1981*; *von Hippel and Berg, 1989*). Consistent with this, yeast SWR1, an INO80 family remodeler that performs histone H2A.Z exchange, searches for nucleosome targets by 1D diffusion on free DNA constrained between reconstituted nucleosomes (*Carcamo et al., 2022*). Although single-nucleosome fluorescence resonance energy transfer studies have indicated that RSC, INO80, ACF, and Chd1, but not SWR1, can utilize ATP hydrolysis to reposition nucleosome core histones on DNA at short length scales (*Blosser et al., 2009*; *Harada et al., 2016*; *Sabantsev et al., 2019*; *Deindl et al., 2013*; *Zhou et al., 2018*; *Qiu et al., 2017*) the processivity and extent of histone octamer movement or 'nucleosome sliding' is unknown and dynamic remodeler engagement and directional nucleosome translocation have not been systematically studied.

Here, we investigate RSC and ISW2 remodeler 1D diffusion using naked DNA and sparsely reconstituted nucleosome array substrates both stretched by optical tweezers to ~5 pN of tension and imaged by confocal microscopy to analyze remodeler–nucleosome interactions. Differences in remodeler diffusion on naked DNA under varying ionic strength and nucleotide conditions indicate that RSC scans DNA mainly by 1D hopping, while ISW2 undergoes 1D sliding. We also observed remodeler–remodeler collisions, and infrequent bypassing events. 1D interactions between a remodeler and a nucleosome are similarly dynamic, with many collisions resulting in confined diffusion or colocalization. In the presence of ATP, both RSC and ISW2 show striking processivity and nucleosome translocation in opposing directions on sparse nucleosome arrays. We discuss these findings in the context of target search and remodeler dynamics which may contribute to NDR expansion or contraction in cells.

## Results

### 1D diffusion of RSC and ISW2 on DNA reveal distinct hopping and sliding modes

DNA helicases and the chromatin remodeler SWR1 have been shown to undergo 1D diffusion on dsDNA (*Carcamo et al., 2022*; *Ramírez Montero et al., 2023*). Given the enrichment of RSC and ISW2 at NDRs and flanking +1 and −1 nucleosomes (*Ng et al., 2002*; *Kubik et al., 2019*), we hypothesized that both remodelers should be capable of free 1D Brownian diffusion on dsDNA (*Figure 1A*). Accordingly, we employed dual optical tweezers and scanning confocal microscopy at 72 nm (225 bp) 1D spatial resolution with laminar flow microfluidics (*Figure 1—figure supplement 1A*) to directly visualize and quantify diffusion of RSC and ISW2 complexes on stretched lambda DNA (48.6 kbp) (Materials and methods). We observed 1D dynamics of purified, functionally active, RSC and ISW2 fused to a HaloTag moiety on the catalytic ATPase subunit (*Figure 1—figure supplement 1B, C*; *Figure 1—figure supplement 2A, B*; Materials and methods). Fluorescent labeling efficiency was high; proteins were purified using standard glycerol gradient centrifugation methods (*Figure 1—figure supplement 2C–F*). In contrast to the stationary dCas9 control (*Sternberg et al., 2014*), both ISW2 and RSC display visible movements on DNA, as evidenced by the overlay of compiled kymographs exhibiting

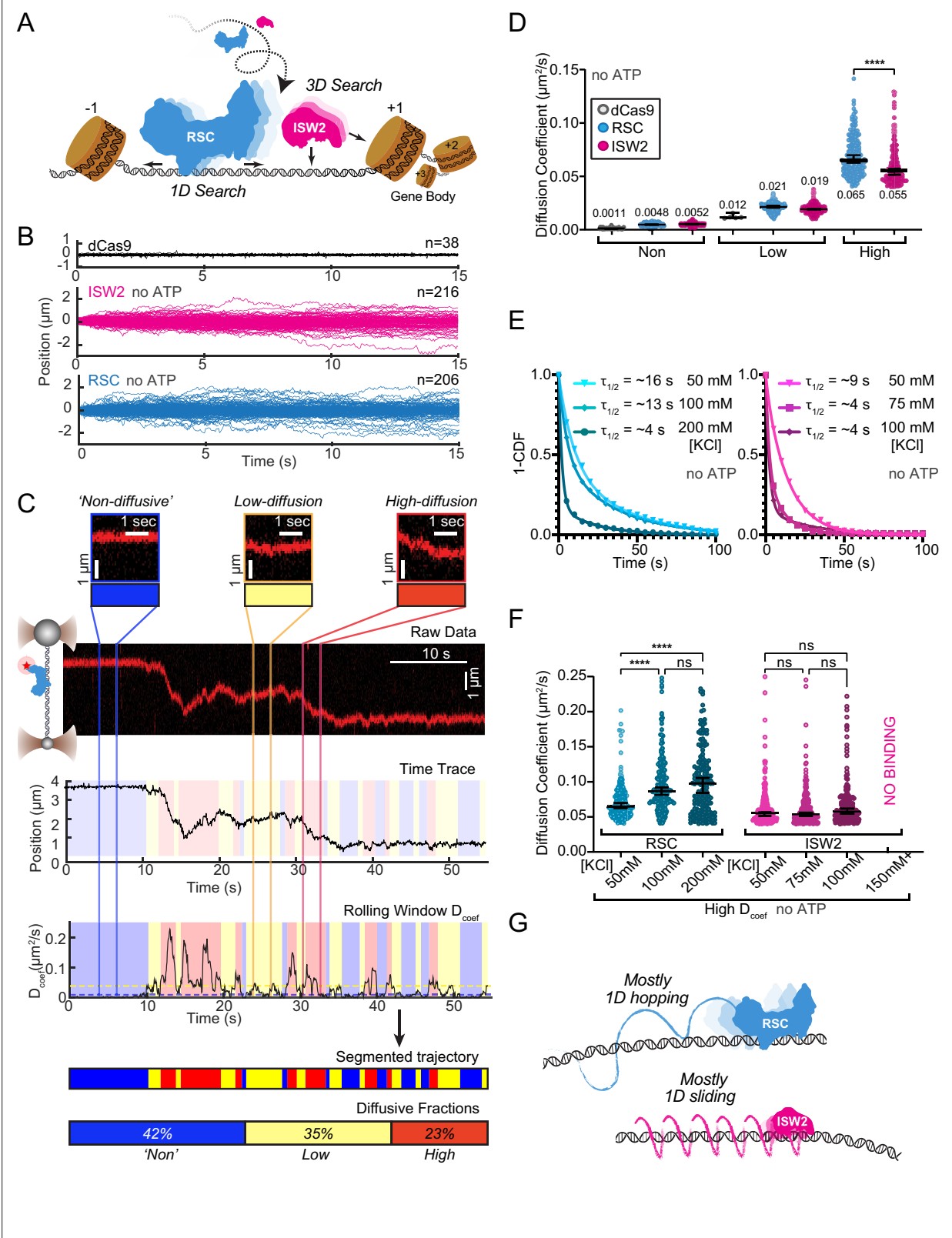

**Figure 1.** RSC and ISW2 diffusion on lambda DNA and impacts of ionic strength. (**A**) Schematic: RSC and ISW2 bind nucleosome-free DNA at yeast promoters. (**B**) Aligned trajectories for RSC, ISW2, and dCas9. (**C**) Rolling-window analysis assigns diffusion coefficients ($D_{coef}$) and percentages for non-diffusive, low, and high diffusion. (**D**) Heterogeneous one-dimensional (1D) diffusion of RSC and ISW2 on lambda DNA (50 mM KCl, no ATP). Scatter violin plots show mean $D_{coef}$ per diffusion type. Mann–Whitney tests compare RSC vs ISW2 distributions in the high-diffusive category, asterisks indicate

*Figure 1 continued*

significance. Median $D_{coef}$ values with 95% confidence intervals are shown. (**E**) Dwell times for RSC and ISW2 at varying KCl concentrations; single-exponential decay fit to the 1-cumulative distribution function (CDF) and half-lives shown. (**F**) Ionic strength impact on high-mobility diffusion at different KCl concentrations. Scatter violin plots depict median values and 95% confidence intervals, asterisks indicate significance. (**G**) Model: RSC primarily undergoes 1D hopping, while ISW2 performs helically coupled 1D sliding.

The online version of this article includes the following source data and figure supplement(s) for figure 1:

**Figure supplement 1.** RSC and ISW2 diffusion analysis under varied conditions.

**Figure supplement 1—source data 1.** Original sodium dodecyl sulfate–polyacrylamide gel electrophoresis (SDS–PAGE) gel showing purified RSC complex.

**Figure supplement 1—source data 2.** Original sodium dodecyl sulfate–polyacrylamide gel electrophoresis (SDS–PAGE) gel showing purified RSC complex with annotation.

**Figure supplement 1—source data 3.** Red channel fluorescence scan of original sodium dodecyl sulfate–polyacrylamide gel electrophoresis (SDS–PAGE) gel showing purified JFX650-labeled RSC complex.

**Figure supplement 1—source data 4.** Red channel fluorescence scan of original sodium dodecyl sulfate–polyacrylamide gel electrophoresis (SDS–PAGE) gel showing purified JFX650-labeled RSC complex with annotation.

**Figure supplement 1—source data 5.** Original sodium dodecyl sulfate–polyacrylamide gel electrophoresis (SDS–PAGE) gel showing purified ISW2 complex.

**Figure supplement 1—source data 6.** Original sodium dodecyl sulfate–polyacrylamide gel electrophoresis (SDS–PAGE) gel showing purified ISW2 complex with annotation.

**Figure supplement 1—source data 7.** Red channel fluorescence scan of original sodium dodecyl sulfate–polyacrylamide gel electrophoresis (SDS–PAGE) gel showing purified JFX650-labeled ISW2 complex.

**Figure supplement 1—source data 8.** Red channel fluorescence scan of original sodium dodecyl sulfate–polyacrylamide gel electrophoresis (SDS–PAGE) gel showing purified JFX650-labeled ISW2 complex with annotation.

**Figure supplement 2.** Halo-tagged remodeler functional validation, labeling, and purification.

**Figure supplement 2—source data 1.** Original native PAGE analysis of nucleosome sliding by HaloTagged RSC.

**Figure supplement 2—source data 2.** Original native PAGE analysis of nucleosome sliding by HaloTagged RSC with annotation.

**Figure supplement 2—source data 3.** Original native PAGE analysis of nucleosome sliding by HaloTagged ISW2.

**Figure supplement 2—source data 4.** Original native PAGE analysis of nucleosome sliding by HaloTagged ISW2 with annotation.

**Figure supplement 2—source data 5.** Original red channel fluorescence scan of a sodium dodecyl sulfate–polyacrylamide gel electrophoresis (SDS–PAGE) gel of JFX650-Halo-RSC used for quantifying Halo-RSC labeling efficiency.

**Figure supplement 2—source data 6.** Original red channel fluorescence scan of a sodium dodecyl sulfate–polyacrylamide gel electrophoresis (SDS–PAGE) gel of JFX650-Halo-RSC used for quantifying Halo-RSC labeling efficiency with annotation.

**Figure supplement 2—source data 7.** Original green channel fluorescence scan of a sodium dodecyl sulfate–polyacrylamide gel electrophoresis (SDS–PAGE) gel of JFX554-Halo-RSC used for quantifying Halo-RSC labeling efficiency.

**Figure supplement 2—source data 8.** Original green channel fluorescence scan of a sodium dodecyl sulfate–polyacrylamide gel electrophoresis (SDS–PAGE) gel of JFX554-Halo-RSC used for quantifying Halo-RSC labeling efficiency with annotation.

**Figure supplement 2—source data 9.** Original image of protein staining of sodium dodecyl sulfate–polyacrylamide gel electrophoresis (SDS–PAGE) gel of JFX 650 and JFX 554-Halo-RSC used for quantifying Halo-RSC labeling efficiency.

**Figure supplement 2—source data 10.** Original image of protein staining of sodium dodecyl sulfate–polyacrylamide gel electrophoresis (SDS–PAGE) gel of JFX 650 and JFX 554-Halo-RSC used for quantifying Halo-RSC labeling efficiency with annotation.

**Figure supplement 2—source data 11.** Original image of flamingo stained SDS–PAGE gel of fractions from glycerol gradient purification for the RSC-JFX650 preparation.

**Figure supplement 2—source data 12.** Original image of flamingo stained sodium dodecyl sulfate–polyacrylamide gel electrophoresis (SDS–PAGE) gel of fractions from glycerol gradient purification for the RSC-JFX650 preparation with annotations.

**Figure supplement 2—source data 13.** Original red channel image of sodium dodecyl sulfate–polyacrylamide gel electrophoresis (SDS–PAGE) gel of fractions from glycerol gradient purification for the RSC-JFX650 preparation.

**Figure supplement 2—source data 14.** Original red channel image of sodium dodecyl sulfate–polyacrylamide gel electrophoresis (SDS–PAGE) gel of fractions from glycerol gradient purification for the RSC-JFX650 preparation with annotations.

**Figure supplement 2—source data 15.** Original image of flamingo stained sodium dodecyl sulfate–polyacrylamide gel electrophoresis (SDS–PAGE) gel of fractions from glycerol gradient purification for the ISW2-JFX650 preparation.

**Figure supplement 2—source data 16.** Original image of flamingo stained sodium dodecyl sulfate–polyacrylamide gel electrophoresis (SDS–PAGE) gel of fractions from glycerol gradient purification for the ISW2-JFX650 preparation with annotations.

*Figure 1 continued*

**Figure supplement 2—source data 17.** Original red channel image of sodium dodecyl sulfate–polyacrylamide gel electrophoresis (SDS–PAGE) gel of fractions from glycerol gradient purification for the ISW2-JFX650 preparation.

**Figure supplement 2—source data 18.** Original red channel image of sodium dodecyl sulfate–polyacrylamide gel electrophoresis (SDS–PAGE) gel of fractions from glycerol gradient purification for the ISW2-JFX650 preparation with annotations.

diffusion away from their initial binding sites (*Figure 1B*). We used 20-frame (0.85 s) rolling windows along the length of the trajectory to quantify instantaneous diffusion coefficients [short-range diffusion] (*Figure 1C*, Materials and methods), classifying trajectory segments into 'non-diffusive', low-diffusive, and high-diffusive groups for RSC (*Figure 1C*) and ISW2 (*Figure 1—figure supplement 1D*).

The 'non-diffusive' group, with diffusion coefficients less than 0.01 µm²/s, is indistinguishable from dCas9 and represents either no diffusion or very slow diffusion below the detection limit (*Figure 1D*). As an important note, directional movements with speeds slower than ~300 bp/s on linear DNA could not be detected using the rolling-window method and were only revealed when detection is extended over a longer duration (e.g., long-range translocation, described in greater detail later). Thus, rolling-window analysis groups short-range translocation and stationary colocalizations in the same 'non-diffusive' category. On average, one-third of time traces are classified as 'non-diffusive' for both remodelers (*Figure 1—figure supplement 1D*). The high-diffusive group, with diffusion coefficients equal to or greater than 0.04 µm²/s, represents the upper-limit diffusion for both RSC and ISW2 (*Figure 1D*). The data show that remodelers undergo frequent transitions during 1D scanning between non-diffusion, low, and high diffusion (*Figure 1C*). These diffusive transitions, including the presence of the intermediate, low-diffusive category, could be due to differences in interaction energies with the underlying DNA sequence and intrinsic remodeler conformations that vary in DNA affinity along the lambda genome sequence (*Visnapuu and Greene, 2009*; *Lorch and Kornberg, 2017*; *Behe, 1995*).

Under the same buffer conditions, RSC is more diffusive than ISW2, as evidenced by its larger diffusion coefficient ($D_{coef}$) (*Figure 1D*) and increased frequency in the high-diffusive category (*Figure 1—figure supplement 1D*). Considering the larger size of RSC (~1 MDa) compared to ISW2 (~300 kDa), our observation is inconsistent with the predicted diffusion based on the Stokes–Einstein equation relating $D_{coef}$ to the particle size, suggesting that the two remodelers utilize different diffusive mechanisms (*von Hippel and Berg, 1989*; *Schurr, 1979*; *Bagchi et al., 2008*; *Ahmadi et al., 2018*). Of the two main types of 1D diffusion, hopping and sliding (*Berg et al., 1981*; *Bonnet et al., 2008*; *Mirny et al., 2009*), 1D hopping, but not 1D sliding, is sensitive to screening ions, and is not constrained to the DNA helical axis. In contrast, 1D sliding is constrained to follow the helical axis and is speed limited by rotational drag (*Blainey et al., 2009*). We measured diffusion for each remodeler under increasing salt conditions and found that both RSC and ISW2 exhibit shorter bound lifetimes at higher salt (*Figure 1E*). Importantly, however, RSC displays elevated high diffusion coefficients (*Figure 1F*, *Figure 1—figure supplement 1E*) and longer durations in the high-diffusive category (*Figure 1—figure supplement 1F*) while these parameters for ISW2 remain unaffected by elevated salt up to 100 mM KCl; binding is lost at 150 mM KCl (*Figure 1F*, *Figure 1—figure supplement 1E, F*). These findings suggest that RSC largely utilizes a 1D hopping mode of diffusion, whereas ISW2 employs helically-coupled sliding (*Figure 1G*). Additionally, we found that RSC displays a substantially higher diffusion with ATPγS, suggesting that conformational changes associated with nucleotide binding may induce a more diffusion-competent state (*Figure 1—figure supplement 1G, H*).

## Remodeler–remodeler collisions during 1D search

Yeast gene promoters are bustling double-helical thoroughfares for transcription and chromatin regulators, the latter displaying macroscopic promoter occupancies of 10–90%, despite rapid on–off kinetics, for any temporal period in live cells, compared to the ~5% occupancies of most general transcription factors (*Kim et al., 2021*; *Ranjan et al., 2020*). This raises the possibility of encounters or collisions in 1D between two or more remodelers on the same piece of accessible DNA. We used differentially labeled RSC and ISW2 to directly visualize heterotypic ISW2–RSC (*Figure 2A*) and homotypic RSC–RSC (*Figure 2B*) collisions, identifying three types of interactions (*Figure 2C–E*). The first, a short-lived colocalization followed by recoil (*Figure 2C*), had half-lives of $t_{1/2}$ 0.033 s for ISW2–RSC and $t_{1/2}$ 0.031 s for RSC–RSC (*Figure 2F, G*). The second type, longer-lived colocalization events involving

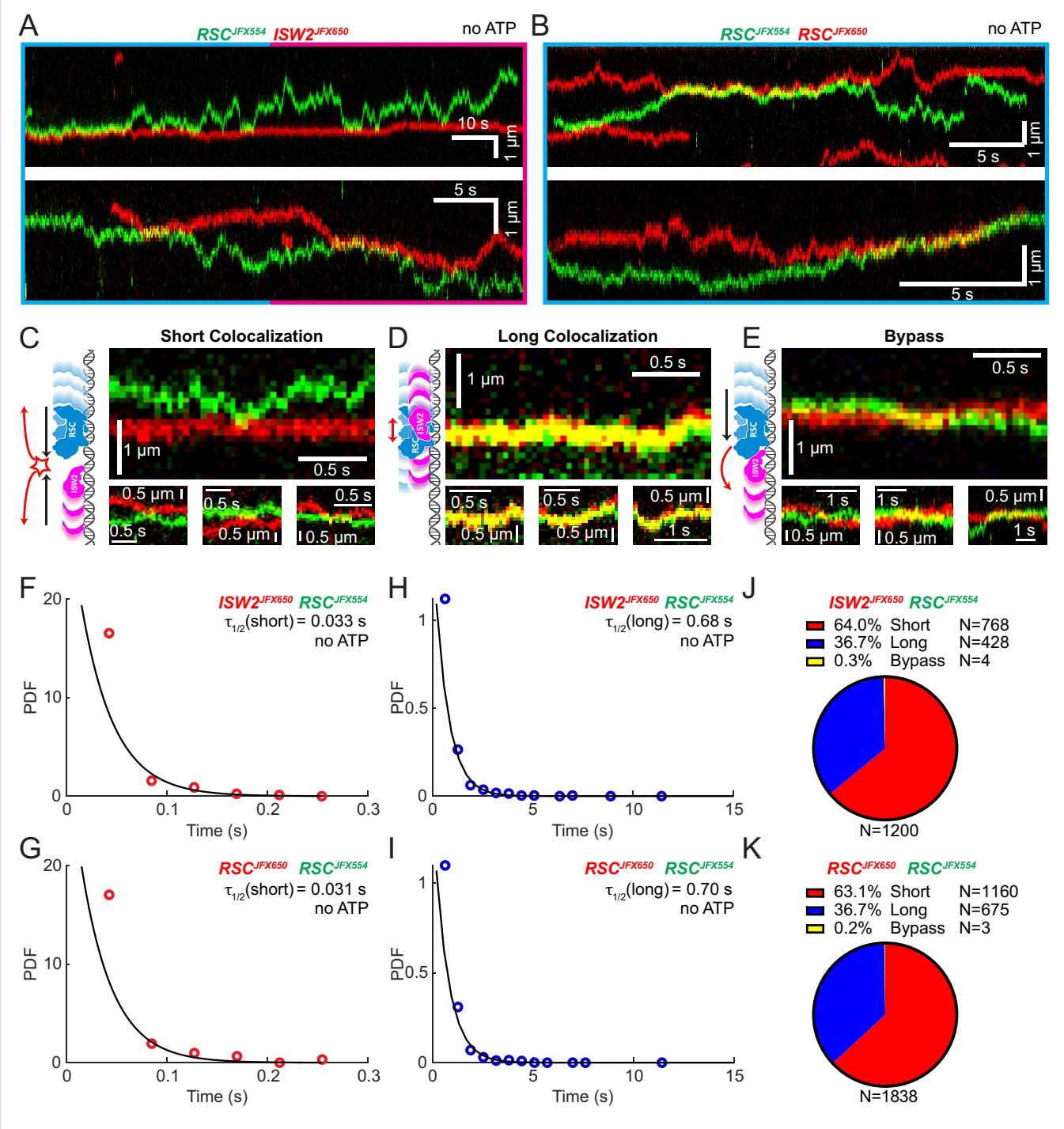

**Figure 2.** Bimolecular remodeler–remodeler interactions during one-dimensional (1D) encounters on DNA. Two-color kymographs of RSC-JFX554 and ISW2-JFX650 (**A**) or RSC-JFX554 and RSC-JFX650 (**B**) diffusing together on naked DNA. Three types of encounter events are observed: short colocalizations (**C**), long colocalizations (**D**), and bypass events (**E**). Representative kymograph sections and corresponding cartoons illustrate each interaction. Dwell times of short colocalization events for RSC–ISW2 (**F**) and RSC–RSC (**G**). Dwell times of long colocalization events for RSC–ISW2 (**H**) and RSC–RSC (**I**). Interaction lifetimes are determined by measuring the time remodelers spend in close proximity. Interaction half-lives ($\tau$) are calculated from single-exponential fits to probability distribution functions (PDF) plots of lifetimes. Proportions of each event type quantified as pie charts for RSC–ISW2 (**J**) and RSC–RSC (**K**).

The online version of this article includes the following figure supplement(s) for figure 2:

**Figure supplement 1.** Detection of remodeler–remodeler interactions and dwell-time estimation.

brief co-diffusion of two remodelers (*Figure 3D*), had half-lives of $t_{1/2}$ 0.68 s for ISW2–RSC and $t_{1/2}$ 0.70 s for RSC–RSC (*Figure 2H, I*). These event types and their durations were confirmed by simulation analysis (*Figure 2—figure supplement 1A–F*). The third type, rare bypass events (*Figure 2E*), constituted only 0.2–0.3% of all colocalizations for both ISW2–RSC and RSC–RSC encounters. The distribution in event types (*Figure 2J, K*) suggests that remodelers act not only as mutual roadblocks but surprisingly, also exhibit mutual affinity upon encounter on DNA, resulting in transient 1D diffusion jointly.

## Nucleosomes halt 1D diffusion and sequester RSC and ISW2

How do remodelers behave upon encountering a nucleosome via 1D diffusion? We investigated remodeler–nucleosome collisions using sparse-density nucleosome arrays reconstituted with site-specifically labeled histone octamers and imaged by pulsed (Cy3) or continuous (JFX554) laser excitation (*Figure 3A–C*; *Figure 3—figure supplement 1A*). We estimated the number of nucleosomes by counting force-induced unwrapping events at 15 pN or higher [25 nm length extension per nucleosome unwrapping] (*Figure 3B*; *Spakman et al., 2020*; *Brower-Toland et al., 2002*; *Ngo et al., 2015*) and collected data on fibers containing ~10 nucleosomes/array (RSC and ISW2) and ~30 nucleosomes/array (RSC) (*Figure 3C*). Independent of nucleotide conditions, we documented several types of nucleosome encounters on the stretched template (*Figure 3—figure supplement 1B*) including collisions with recoil (*Figure 3—figure supplement 1C*), stable colocalization (*Figure 3—figure supplement 1D*), and nucleosome bypass (*Figure 3—figure supplement 1E*). Only in the presence of ATP, did we observed translocation events, to be further described in the following section (*Figure 3—figure supplement 1F*). In some examples, all event types are observed within a single temporal trace (*Figure 3D, E*). Importantly, owing to low nucleosome density, direct 3D encounter with a nucleosome was not observed, and because protein–DNA interactions are mostly equilibrated in the timeframe (~1 min) between protein introduction, and image acquisition, few new binding events are observed.

As noted earlier, in addition to high and low levels of diffusion, remodelers frequently transition into a 'non-diffusive' state [static sequestration or undetectable movements] of variable duration with a half-life on the order of 3–5 s on naked linear DNA (RSC; $T$ = 3.8 s no ATP, $T$ = 3.3 s ATPγS and ISW2; $T$ = 5.3 s no ATP, $T$ = 3.5 s ATPγS) (*Figure 3F–I*). However, upon nucleosome encounter by either RSC or ISW2, non-diffusive remodeler dwell times increase substantially (fivefold with no ATP; ~eightfold with ATPγS) (RSC; $T$ = 18 s no ATP, $T$ = 28 s ATPγS) (*Figure 3F, J*) (ISW2; $T$ = 28 s no ATP, $T$ = 28 s ATPγS) (*Figure 3G, K*). Of note, in the presence of hydrolysable ATP, we found little substantive change for RSC 'non-diffusive' dwell-time ($T$ = 23 s, ATP), whereas the 'non-diffusive' dwell-time for ISW2 decreased by ~twofold ($T$ = 12 s, ATP) (*Figure 3F, G, J, K*). This could be due to the binding turnover reported for ISW2 in biochemical assays (*Fitzgerald et al., 2004*). Irrespective, we conclude that encounters in 1D between diffusing remodelers and static nucleosomes can result in remodeler sequestration, in all nucleotide conditions. The dwell-time survival plots are best fit to a double exponential decay, with two components showing long and short half-lives of e.g. ~20 and ~2 s for RSC. The basis for a fast-decay component is unclear, but it might be due to a pre-engagement state where RSC is sampling the nucleosome for a more stable interaction.

## ATP-dependent, processive, and directional nucleosome translocation by RSC and ISW2

Since the above short-range analysis of 1D diffusion through the automated rolling-window mean squared displacement (MSD) analysis masks potential ATP-dependent nucleosome translocations within the non-diffusive category (*Figure 3D, E*), we visually screened for long-range nucleosome movements >300 bp (>5 s) over the entire period of fluorophore detection before photobleaching (up to several minutes). Accordingly, we observed numerous ATP hydrolysis-dependent nucleosome translocation events comprising 42% of RSC–nucleosome and 21% of ISW2–nucleosome encounters (*Figure 4A–D*). The relatively low yield of observed translocations may be due to short-range translocation events which may go undetected, and may also be caused by experimental variability including nucleosome and enzyme preparations. Finally, we caution that our data were collected using nucleosomes under tension (~5 pN) which may have helped reveal an activity otherwise not observable without tension.

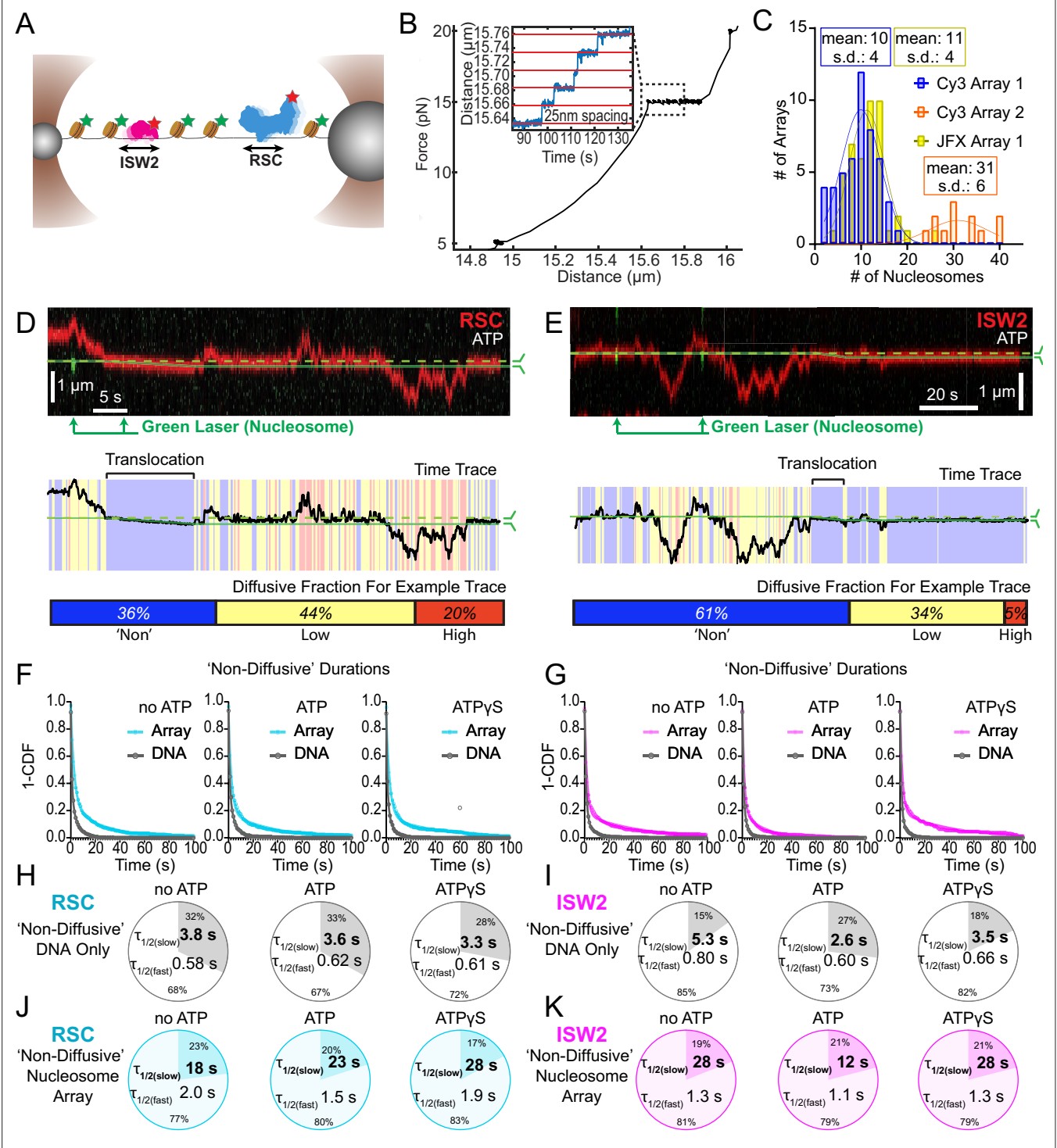

**Figure 3.** Nucleosomes constrain one-dimensional (1D) diffusion of RSC and ISW2. (**A**) Optical tweezers and confocal microscopy assay for measuring remodeler diffusion on lambda DNA nucleosome arrays. (**B**) Force distance plots are used to assess the number of nucleosomes on arrays, with ~25 nm lengthening per nucleosome at 15 pN (inset). (**C**) Histograms show nucleosome numbers per array for three arrays used, with mean values and standard deviations. Kymograph segments of RSC (**D**) and ISW2 (**E**) encountering a nucleosome on the array with 1 mM ATP. Diffusion levels are binned on time trace plots. As a note, nucleosome translocation occurs in each example with the original position of the nucleosome marked by the dashed green line and the changing nucleosome position is marked by the solid green line. Translocation events fall into the 'non-diffusive' category due to their slow motion. Survival plots of dwell times in the 'non-diffusive' state for RSC (**F**) and ISW2 (**G**) on naked DNA or nucleosome array. (**H–K**) Half-lives are

*Figure 3 continued on next page*

*Figure 3 continued*

determined using double exponential fits and are indicated for the slow and fast percentages (pie charts). Values are given for various ATP conditions and for diffusion on naked DNA (RSC; H-ISW2; II) and nucleosome array (RSC; J-ISW2; **K**).

The online version of this article includes the following source data and figure supplement(s) for figure 3:

**Figure supplement 1.** Remodeler–nucleosome interactions and nucleosome sliding assays.

**Figure supplement 1—source data 1.** Original DNA scan of electrophoretic mobility shift assay of lambda nucleosome arrays with increasing octamer concentration.

**Figure supplement 1—source data 2.** Original DNA scan of electrophoretic mobility shift assay of lambda nucleosome arrays with increasing octamer concentration with annotation.

**Figure supplement 1—source data 3.** Original green channel scan of electrophoretic mobility shift assay of lambda nucleosome arrays with increasing octamer concentration.

**Figure supplement 1—source data 4.** Original green channel scan of electrophoretic mobility shift assay of lambda nucleosome arrays with increasing octamer concentration with annotation.

**Figure supplement 2.** Nucleosome translocation durations and half-lives.

Strikingly, nucleosome translocations are very processive, as evidenced by the co-mobility of nucleosome and remodeler fluorescence in one direction for extended periods lasting up to minutes (*Figure 4C, D*). For both RSC and ISW2, translocation events are heterogeneous in their characteristics (*Figure 4E, F*). Translocation events can be of constant speed or of changing speed, they can move in one direction exclusively or exhibit a limited number of direction changes and, while rare, some events are discontinuous, interrupted by nucleosome disengagement with 1D diffusion on DNA (*Figure 4E*). RSC translocation events switch direction 9% of the time (65 traces), but none of the time for ISW2 traces (46 traces).

We calculated the average translocation speeds for both RSC and ISW2. Of all translocating traces, 12% of RSC and 9% of ISW2 traces showed speed changes within the same single-molecule trajectory (*Figure 4G*); the basis for these changes is unclear. To determine translocation speeds for both remodelers, we segmented time traces into single-speed constituents for all translocation events (*Figure 4H*). We also measured the translocation distance in base pairs (*Figure 4I*) and time in seconds (*Figure 4J*) for each single-speed segment. Uninterrupted, speed-constant RSC translocations show a median duration of 33 s (940 bp) whereas ISW2 events were slightly shorter-lived, lasting 22 s (690 bp). Also of note, within a single DNA molecule, translocation events move up and down the nucleosome array with roughly equal frequencies (*Figure 4K*). Interestingly, ISW2 and RSC exhibited the same average translocation speed of 29 bp/s (RSC s.d. 9.8; ISW2 s.d. 14); translocation is ATP-dependent as translocation was not observed in the absence of ATP or in the presence of ATPγS (*Figure 4L*).

In the presence of ATP, for RSC, 28% of 'non-diffusive' times are periods of visible translocation whereas for ISW2 this value is only 14% (these percentages were extracted from the non-diffusive [*Figure 3J, K*] and translocating half-lives [*Figure 4J*]). The lower value for ISW2 could be caused by the known ATP hydrolysis-driven nucleosome-binding turnover. Furthermore, half-lives of translocation events (20 s for RSC and 17 s for ISW2) are comparable to remodeler 'non-diffusive' half-lives on nucleosome arrays (*Figure 3—figure supplement 2A, B*). Thus, the inclusion of translocation events into the non-diffusive category did not impact our reported effects of ATP on nucleosome-bound half-lives.

## RSC pushes whereas ISW2 pulls nucleosomes relative to 1D search on DNA

Both RSC and ISW2 remodelers, acting on promoter nucleosomes, have competing effects on +1 nucleosome positioning in vivo (*Kubik et al., 2019*). RSC widens the NDR by pushing the +1 nucleosome further downstream, while ISW2 narrows the NDR by pulling it in the opposite direction (*Kubik et al., 2019*). We sought to directly visualize nucleosome pushing and pulling relative to 1D remodeler search on DNA. Traces containing both 1D diffusion and translocation events can be used for addressing this question (*Figure 5A, B*). This analysis showed a higher frequency of observed ATP-dependent nucleosome translocation events at the start of RSC imaging as compared to ISW2 (*Figure 5A, B*; blue bars). Given the time lag between remodeler introduction and image acquisition, the higher frequency for RSC may be due to its higher 1D diffusivity. Alternatively, ISW2 is often found

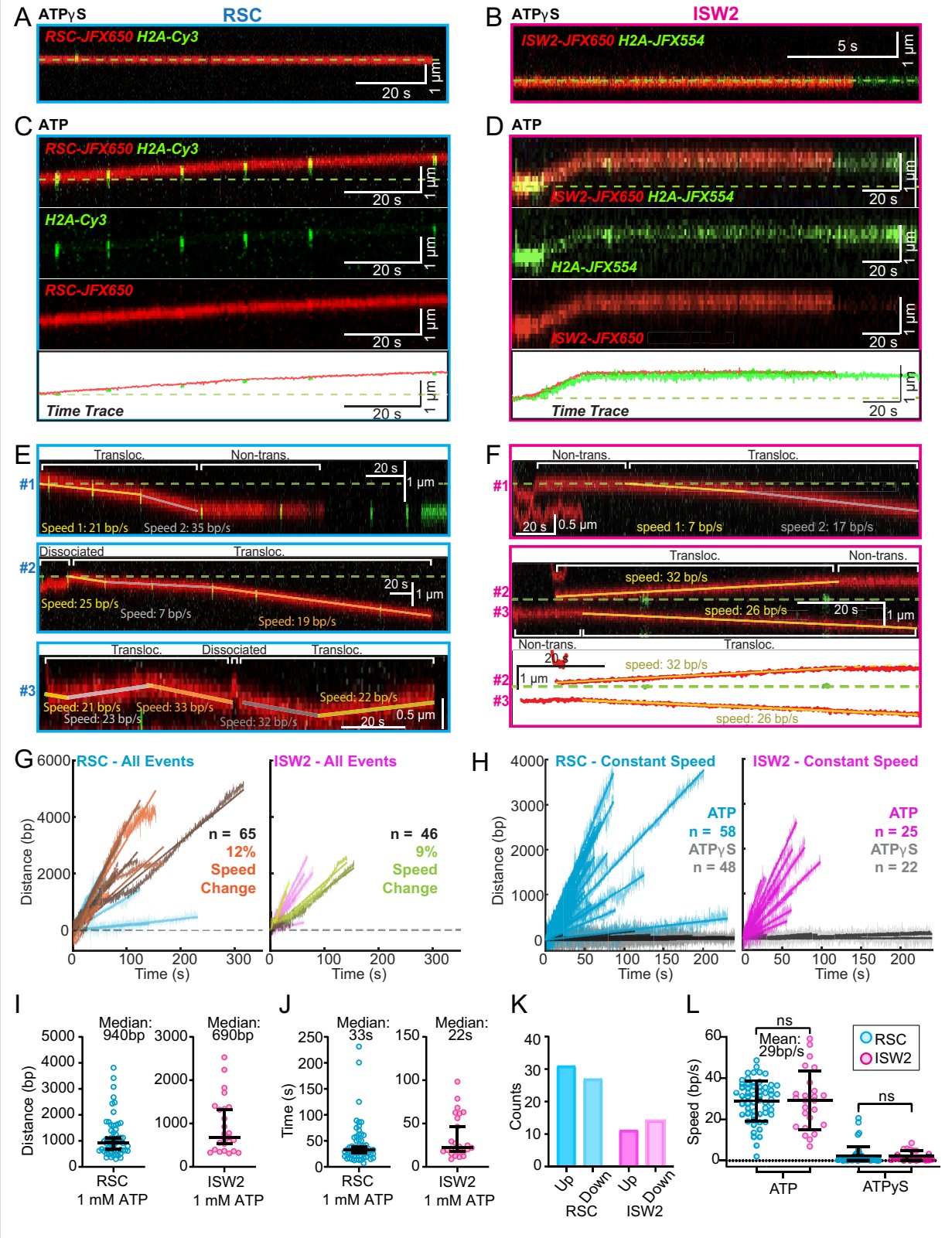

**Figure 4.** ATP-dependent processive nucleosome translocation by RSC and ISW2. Kymographs of RSC (**A**) and ISW2 (**B**) showing static colocalization in 1 mM ATPγS. Kymographs of RSC (**C**) and ISW2 (**D**) moving directionally with nucleosome signals in 1 mM ATP. The green laser is either blinked or continuous, and the initial position of the nucleosome is marked with a dashed green line. Additional examples of nucleosome translocation events for (**E**) RSC and (**F**) ISW2. Examples include cases with speed or direction changes and cases where speed remains constant. Speed changes

*Figure 4 continued on next page*

*Figure 4 continued*

or pauses are indicated by a dashed continuation of the translocation event line. Special events like bypass or brief periods of one-dimensional (1D) search are annotated. Speed values are directly displayed on the kymograph. An example of RSC translocation changing direction is provided, along with a kymograph showing two ISW2 translocation events in opposite directions with an immobile nucleosome as a reference. (**G**) Linear fit to all translocation events that are moving in the same direction, highlighting speed changes. (**H**) Linear fits to constant-speed segments (>300 bp, >5 s, $r^2 >$ 0.5), with ATPγS static colocalization controls in gray. (**I**) Scatter plot of distance (base pairs) covered by single-speed segments. (**J**) Scatter plot of time duration (seconds) of single-speed segments. (**K**) Directionality of translocation with respect to the tether orientation indicating random preference for translocation direction on DNA when not accounting for the 1D approach direction. (**L**) Scatter plots of translocation speeds for RSC (cyan) and ISW2 (magenta), including mean values and standard deviation error bars; ATPγS control shown alongside the hydrolysable ATP condition.

in 1D diffusive search mode at the start of imaging due to its faster ATP-dependent turnover after nucleosome encounter (*Figure 3K*). We also note that translocation events often terminate in stable, non-diffusive colocalization of remodeler with nucleosomes (*Figure 5A, B*, red bars) rather than loss of remodeler or nucleosome fluorescence (histone ejection).

For the limited subset of RSC and ISW2 kymographs that display both 1D diffusion and translocation, we considered four scenarios: (1) 1D search leading directly to translocation; (2) translocation leading directly to 1D disengagement; (3) 1D search leading to stable non-translocation engagement followed by translocation; and (4) the reverse order of the third scenario (*Figure 5—figure supplement 1A–D*). The results reveal a distinct bias in the direction of translocation relative to 1D diffusion (*Figure 5C–F*). RSC shows a preference for translocation in the same direction as diffusive approach to the nucleosome or translocation in the opposite direction of diffusive disengagement from the nucleosome (20/25 events) (*Figure 5G*). Conversely, ISW2 exhibits a bias for translocation in the opposite direction to its approach (19/21 events) (*Figure 5H*). While more data are required for a conclusive statement, the observed bias provides the first direct visual evidence in support of opposing directions of nucleosome translocation that is a central feature of the push–pull model for expansion and contraction of the NDR (*Figure 5I*).

## Discussion

The historical 'facilitated diffusion' model for how DNA-binding proteins find their targets at 'faster-than-[3D] diffusion-controlled rates' demonstrated the importance of 1D diffusion on noncognate DNA to facilitate target binding (*Berg et al., 1981*). In this context, we investigated the search processes of two chromatin remodelers RSC and ISW2, which target a subset of nucleosomes flanking the short stretches of NDRs at promoters genome-wide. We directly visualized 1D target search by RSC and ISW2 on extended naked DNA within a sparsely reconstituted nucleosome array as a surrogate for yeast NDRs. The two remodelers exhibit different modes of 1D scanning, as shown by their salt sensitivity. ISW2 and RSC favor 1D sliding and 1D hopping, respectively, during their scans of naked DNA.

Despite being depleted of nucleosomes, NDRs at promoter regions resemble congested marketplaces with active protein traffic from multiple transcription components and regulators. Live-cell, single-particle imaging studies have demonstrated a likelihood of two yeast remodelers co-occupying the same promoter, raising the possibility of interference during the target search process (*Kim et al., 2021*). For the bacterial Lac operon, it has been previously theorized that molecular crowding limits diffusion to the 'vacancies' between LacI and other proteins bound to DNA (*Li et al., 2009*). Supporting this concept, our data show that RSC and ISW2, mutually confine their individual diffusion, and at times even diffuse together briefly, suggestive of remodeler–remodeler interactions promoted by heretofore undocumented DNA-guided association. These collisions and colocalizations are highly transient, on a sub-second timescale, and 1D diffusion is also very fast, with RSC and ISW2 able to individually scan 150 bp of free DNA within 0.2 s. Thus, the 1D search time after stochastic binding to the NDR constitutes only a small fraction of the overall in vivo remodeler lifetime (several seconds) bound to chromatin (*Kim et al., 2021*). These kinetics of remodeler 1D diffusion on naked NDRs are unlikely to be functionally rate-limiting, which underscores the regulatory potential of diffusion interference by nonhistone barriers, in addition to potential controls on 3D nucleoplasmic diffusion and remodeler recruitment to the NDR, oriented nucleosome engagement, and activation of the bound catalytic ATPase motor.

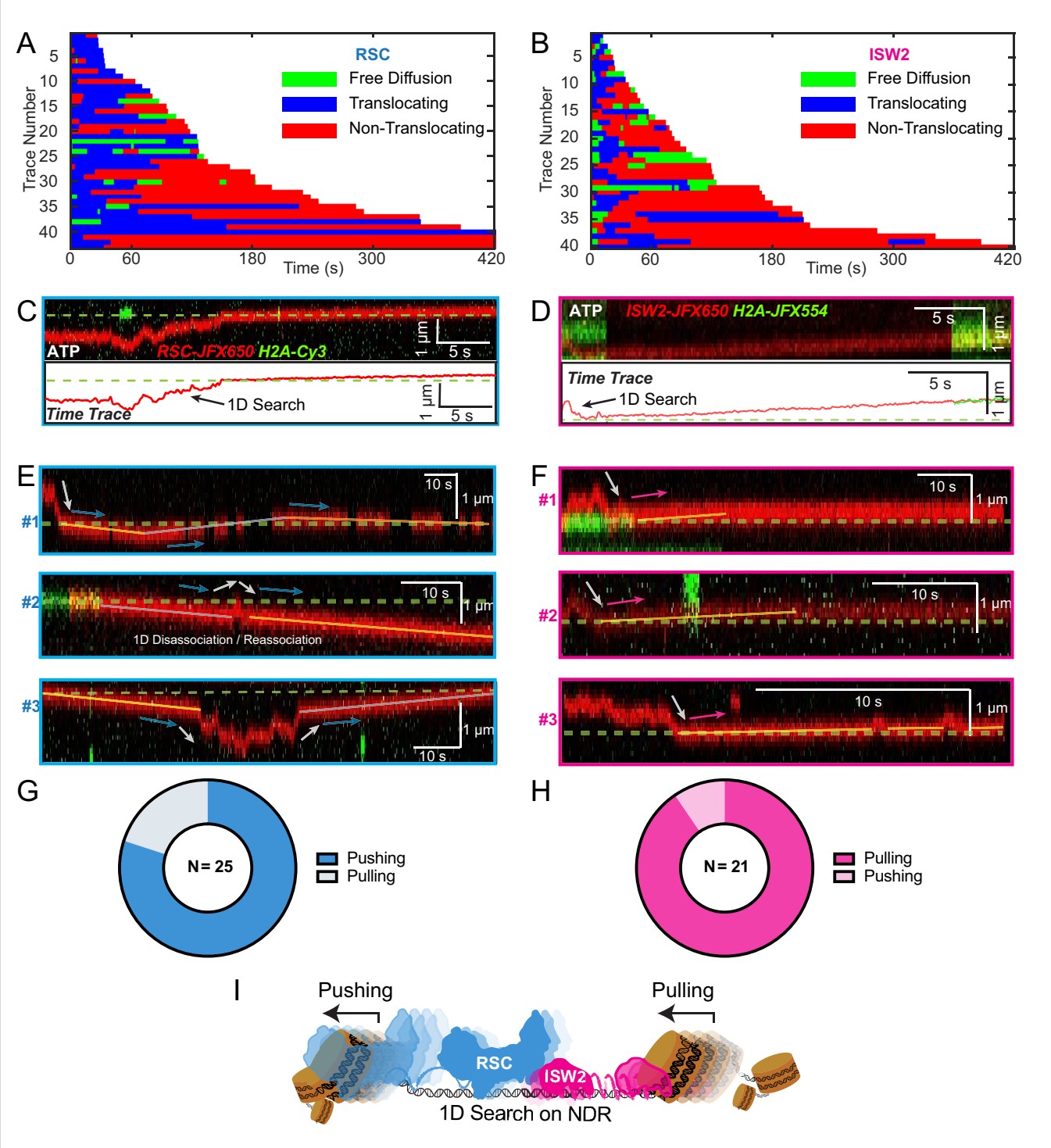

**Figure 5.** Translocation direction relative to one-dimensional (1D) diffusion on linker DNA supports RSC and ISW2 'push–pull' models. Trace fragmentation plots illustrate translocating regardless of changes in speed or changes in direction (blue), non-translocating (red), and free diffusion (green) segments for RSC (**A**) and ISW2 (**B**). Examples of translocation events following 1D encounter for RSC (**C**) and ISW2 (**D**) shown alongside single-particle traces for clarity. (**E, F**) Additional examples of RSC and ISW2 encountering a nucleosome through 1D diffusion and directionally translocating it. Arrows indicate the direction of the remodeler's approach and subsequent nucleosome translocation. Pie charts of 'pushing' and 'pulling' counts for RSC (**G**) and ISW2 (**H**); all types of observations considered. (**I**) Summary schematic.

The online version of this article includes the following figure supplement(s) for figure 5:

**Figure supplement 1.** Various translocation scenarios support the push–pull model for RSC–ISW2.

Notably, we directly captured long-range processive translocation of individual nucleosomes by both RSC and ISW2 over kbp distances at speeds of 29 bp/s. Previous studies established remodeler-driven nucleosome translocations on short DNA fragments (~200 bp) (*Blosser et al., 2009*; *Harada et al., 2016*; *Sabantsev et al., 2019*; *Fitzgerald et al., 2004*; *Hota et al., 2013*; *Deindl et al., 2013*). These studies revealed that RSC (*Harada et al., 2016*) and ISW2 (*Hota et al., 2013*; *Deindl et al., 2013*) generate unidirectional nucleosome translocation events, whereas ISW1, CHD1, and the Isw2p catalytic subunit of ISW2 remodelers produce back and forth nucleosome translocations (*Blosser et al., 2009*; *Qiu et al., 2017*; *Deindl et al., 2013*), at speeds of ~2 bp/s (*Blosser et al., 2009*). Although short-range ATP-dependent nucleosome translocations are not resolved by our instantaneous diffusion analysis limited to a 1D localization precision of 226 ± 32 bp (72 ± 10 nm), our long-range analysis demonstrates speeds >10-fold higher than previously reported for ACF (*Blosser et al., 2009*). This difference could be due to sub-saturating concentrations of ATP used in previous studies to resolve single base-pair stepping (*Blosser et al., 2009*; *Harada et al., 2016*; *Sabantsev et al., 2019*; *Fitzgerald et al., 2004*; *Hota et al., 2013*; *Deindl et al., 2013*), to the sequence of lambda DNA rather than the high-affinity 601-nucleosome positioning sequence, and to substrate tension (5 pN) introduced by stretching a sparsely reconstituted nucleosome array between two optical traps in the force regime where there is unwrapping of nucleosomal DNA (*Brower-Toland et al., 2002*; *Ngo et al., 2015*; *Mihardja et al., 2006*). The nucleosome array or free DNA is stretched to 5 pN of tension prior to image acquisition, and during imaging, tension is allowed to fluctuate freely. These tensions may be causative of the heterogeneous, long-range processive translocation behavior we have observed. Of interest, a study previously showed that RSC associates in vivo with partially unwrapped nucleosomes at MNase sensitive promoters (*Brahma and Henikoff, 2019*). Partially unwrapped nucleosomes could be optimal substrates for both RSC- and ISW2-mediated nucleosome sliding. Although chromatin fiber compaction helps stabilize nucleosomes from torsional stress arising from transcription in cells (*Kaczmarczyk et al., 2020*), tension-induced unwrapping of nucleosomes may also occur and help transiently stimulate similar levels of processivity and speed to that which we report. Moreover, in cells, nucleosome arrays are closely spaced, and the translocating nucleosome may be blocked by the presence of downstream arrays. Further investigations are necessary to dissect the underlying causes of remodeler-induced nucleosome translocation under DNA tension and assess the influence of periodically spaced nucleosome arrays.

It is of interest that histone H2A fluorescence is retained over long-lived nucleosome translocation events, indicating the absence of nucleosomal H2A dissociation or eviction despite tension-induced DNA unwrapping. This is anticipated for ISW2, as the ISWI family repositions nucleosomes without histone eviction (*Hamiche et al., 1999*; *Längst et al., 1999*), but surprising for RSC which has been found to evict histone octamers (*Rawal et al., 2018*). However, our assays are conducted in the absence of histone chaperone Nap1 which is known to facilitate histone eviction (*Prasad et al., 2016*). The influence of histone chaperones on processive nucleosome translocation and histone eviction is also a topic for future study.

While it should be taken as a caveat that all our assays were performed under approximately 5 pN of tension, the observations of long-range translocation permitted the interrogation of the directionality of translocation events relative to 1D search, an opportunity not available under the experimental conditions of short-range translocation. RSC and ISW2 have opposing effects on positioning NDR-flanking nucleosomes in vivo, yet it has been unclear whether the two remodelers possess intrinsically opposed directionality relative to the NDR during nucleosome translocation to either expand or contract NDR size (*Donovan et al., 2021*; *Oberbeckmann et al., 2021*; *Chen et al., 2022*). Our findings of bias for translocation by RSC in the same direction as approach to the nucleosome, and bias for translocation by ISW2 in the opposite direction provides direct evidence in support of the 'push–pull' mechanism for controlling NDR width. Thus, 1D hopping by RSC on naked DNA between sparse nucleosome arrays, nucleosome engagement and translocation in the same direction as its approach would expand the NDR (pushing away from the long nucleosome linker) while 1D sliding and nucleosome translocation by ISW2 in the opposite direction shrinks the NDR (pulling toward the long nucleosome linker). In the context of the native NDR, we speculate that, once bound, remodelers rapidly traverse the NDR by 1D sliding or hopping to engage with and translocate +1 and −1 nucleosomes until halted by upstream and downstream nucleosomes or other protein roadblocks.

The direction of nucleosome translocation appears to be biased in a way that is highly relevant to the regulation of NDR size (*Figure 5I*).

Indeed, the recent cryoEM structures of RSC (*Wagner et al., 2020*; *Ye et al., 2019*) demonstrating asymmetric orientation of RSC DNA-binding components on a long-linker nucleosome relative to ATPase contact at SHL-2 of the core particle provides the structural basis for movement of +1 and −1 nucleosomes away from the NDR. Although similar structural data for ISW2 is unavailable, a proposed model based on biochemical and footprinting analyses predicts ISW2-driven nucleosome movement in the direction toward the NDR (*Hota et al., 2013*). Additional mechanisms that may contribute to directional nucleosome movement include promoter DNA sequence-binding preferences of RSC to GC-rich motifs and poly dA–dT stretches that may orient the remodeler on the NDR (*Badis et al., 2008*), the introduction of diffusion barriers in the NDR by stable, high-occupancy 'pioneering' transcription factors (*Chen et al., 2022*; *Donovan et al., 2023*; *Krietenstein et al., 2016*; *Mivelaz et al., 2020*), and sensing of mechanically rigid DNA in the NDR as was shown for INO80 (*Basu et al., 2021*). The convergence of single-molecule imaging, structural, biochemical, and genomic approaches offers deep insights and exciting opportunities for understanding the biogenesis and maintenance of chromatin accessibility that is fundamental to the regulation of genome-based activities.

## Materials and methods
### Dual optical tweezers and confocal microscopy
Imaging was performed using a commercial optical tweezer combined with fluorescence microscope, C-Trap (LUMICKS, Amsterdam). A laminar flow-based microfluidics chamber was used for data collection, and reproducible measurements of remodeler diffusion were made in the absence of flow in the channel 4 protein reservoir (*Figure 1—figure supplement 1A*). The microfluidics system was passivated before imaging: bovine serum albumin (BSA; 0.1% wt/vol in phosphate-buffered saline [PBS]) and Pluoronics F128 (0.5% wt/vol in PBS) were each flowed for 30 min, followed by a 30 min flush with PBS. Bacteriophage $\lambda$ DNA, dual labeled with 3x-Biotin at one end and 3x-Digoxigenin at the other, was tethered between 4.38 μm SPHERO Streptavidin Coated polystyrene beads (Spherotech, Cat. No. SVP-40-5) in Trap1 and 2.12 μm SPHERO Anti-Dig polystyrene beads (Spherotech, Cat. No. DIGP-20-2) in Trap2. The trapping laser was set to 100% and overall power to 30%, with a Trap 1 split power of 60%. The tethered $\lambda$ DNA was stretched to 5 pN force, and trap positions were fixed during kymograph acquisition.

Either 500 pM or 1 nM RSC$^{Sth1-3xFlag-Halo}$, or 136.6 pM ISW2$^{Isw2-3xFlag-Halo}$ in imaging buffer, were flowed at low pressure. Imaging buffer consisted of saturated Trolox solution and RSC reaction buffer (10 mM Tris, pH 7.4, 50 mM KCl, 3 mM MgCl$_2$, 0.1 mg/ml BSA). The same imaging buffer was used for both RSC and ISW2. For confocal microscopy, a 42.4-ms line time, 6.5% red laser power, and 15% green laser power were utilized. Emission filters for blue (500/525 nm), green (545/620 nm), and red (650/750 nm) lasers were employed.

### Purification and labeling of chromatin remodeling complexes
Catalytic subunit genes Sth1 and Isw2 were C-terminally tagged with 3xFlag-HaloTag at their endogenous loci in *S. cerevisiae* for native remodeler complex purification (*Figure 1—figure supplement 1B*). Yeast cultures (4 l) were grown in YPAD medium (1% Bacto yeast extract, 2% Bacto peptone, 3% dextrose, 0.004% Adenine hemisulfate) to an optical density (OD600) of 3.5–4.0. Cells were harvested by centrifugation, washed twice in water, once in resuspension buffer (200 mM 4-(2-hydroxyethyl)-1-piperazineethanesulfonic acid) (HEPES), 1 mM ethylenediaminetetraacetic acid (EDTA), 40% glycerol, 100 mM KOAc, 0.284 μg ml$^{-1}$ leupeptin, 1.37 μg ml$^{-1}$ pepstatin A, 0.17 mg ml$^{-1}$ phenylmethylsulfonyl fluoride (PMSF), 0.33 mg ml$^{-1}$ benzamidine). The cell pellet was flash-frozen in liquid nitrogen. Lysis was achieved by cryo-milling (Spex Freezer/Mill 6870) for 15 cycles with alternating 1 min 'on' and 1 min rest periods. The resulting powder was resuspended in approximately half of the powder volume of lysis buffer (150 mM HEPES pH 7.6, 1 mM EDTA, 2 mM MgCl$_2$, 20% glycerol, 100 mM KOAc, 5 mM β-mercaptoethanol, 0.284 μg ml$^{-1}$ leupeptin, 1.37 μg ml$^{-1}$ pepstatin A, 0.17 mg ml$^{-1}$ PMSF, 0.33 mg ml$^{-1}$ benzamidine, 0.5 mM NaF, 5 mM β-glycerophosphate). Protein extraction was carried out with the addition of 0.3 M KCl and incubated at 4°C for 30 min. The extract was cleared by

centrifugation at 25,000 rpm at 4°C for 2 hr, and the supernatant was incubated with pre-equilibrated 1 ml anti-FLAG M2 agarose (Sigma-Aldrich) for 4 hr at 4°C with gentle rotation.

The agarose resin was washed five times with high-salt wash buffer (20 mM HEPES pH 7.6, 0.2 mM EDTA, 10% glycerol, 500 mM KOAc, 0.01% IGEPAL CA-630, 0.284 µg ml$^{-1}$ leupeptin, 1.37 µg ml$^{-1}$ pepstatin A, 0.17 mg ml$^{-1}$ PMSF, 0.33 mg ml$^{-1}$ benzamidine) and three times with regular salt wash buffer (20 mM HEPES pH 7.6, 0.2 mM EDTA, 10% glycerol, 300 mM KOAc, 0.01% IGEPAL CA-630, 0.284 µg ml$^{-1}$ leupeptin, 1.37 µg ml$^{-1}$ pepstatin A, 0.17 mg ml$^{-1}$ PMSF, 0.33 mg ml$^{-1}$ benzamidine). The bead-bound remodeler complex was eluted twice using 0.5 mg/ml 3x Flag peptide (APExBIO), with the first elution incubated for at least 2 hr. The eluate was concentrated using a 100-kDa molecular weight cut-off (MWCO) centricon column (EMD Millipore), flash-frozen, and stored at −80°C until further processing.

For fluorescent labeling, the eluate was thawed on ice, incubated with 2 µM JFX650 or JFX554 (sourced from Luke Lavis) for 2 hr at 4°C with gentle shaking. The eluate was then applied to a 20–60% glycerol gradient in gradient buffer (25 mM HEPES–KOH pH 7.6, 1 mM EDTA, 2 mM MgCl$_2$, 0.01% NP-40, 300 mM KOAc) for velocity sedimentation, allowing further purification of complexes and separation from unbound free dyes. Centrifugation was performed at 45,000 rpm for 20 hr at 4°C in an SW 60 T rotor. Peak fractions were analyzed by sodium dodecyl sulfate–polyacrylamide gel electrophoresis (SDS–PAGE) and Flamingo Fluorescent Stain (Bio-Rad) (***Figure 1—figure supplement 1C***). Protein concentration was determined by comparing to a serially diluted BSA standard in SDS–PAGE. Remodeler labeling efficiency was deemed to be high based on comparison to protein labeled with large excesses of dye (***Figure 1—figure supplement 2C***).

## Gel-based nucleosome sliding assay

RSC nucleosome sliding reactions were performed using 20 nM nucleosome [43N43, Cy5DNA], 10 nM RSC$^{Sth1-3F-Halo}$, and 1 mM ATP in a buffer containing 10 mM Tris (pH 7.4), 50 mM KCl, 3 mM MgCl$_2$, and 0.1 mg/ml BSA (***Schlichter et al., 2020***). ISW2 nucleosome sliding reactions were conducted with 33.3 nM nucleosome [80N3, Cy5-DNA, Cy3-H2A], 6.15 nM ISW2$^{Isw2-3F-Halo}$, and 1 mM ATP in a buffer composed of 25 mM HEPES–KOH (pH 7.6), 50 mM KCl, 5 mM MgCl$_2$, 0.1 mg/ml BSA, and 5% glycerol. All reactions had a 10-µl total volume and were incubated at 30°C with gentle mixing. At designated time points, reactions were quenched by adding 3 µg of salmon sperm DNA and 5 mM EDTA. Samples were subsequently loaded onto 4.5% or 6% native polyacrylamide gels, and the gels were imaged using Cy5 excitation on a Typhoon Imager System (***Figure 1—figure supplement 2A, B***).

## Single-molecule tracking and analysis

Kymographs were analyzed using the Pylake KymoTracker widget (Lumicks.pylake, ver. 0.10.0) in Python. The following parameters were applied for particle tracking: line width: 0.4 µm, minimum length: 8 pixels, pixel threshold (minimum pixel intensity): 3, window (maximum frames of gap allowed to connect two lines as one track): 8, sigma (fluctuation in the molecule's position over time): 0.14, velocity: 0.00, Refine lines: yes. For tracks with gaps that could not be connected by the default parameters (more commonly observed for long-lived tracks), the 'Connect line' function was used to manually connect two lines. Notably, all movies were acquired without laminar flow, resulting in rare protein-binding events during the movie. Sigma and/or velocity parameters were increased to track highly diffusive molecules.

## Rolling-window diffusion analysis pipeline

To capture transitions in diffusive states, 20-frame window sub-trajectories were made from the beginning to the end of each trajectory. The 20-frame window size was determined by trial-and-error, with shorter windows (e.g., 5, 10 frames) resulting in noisy detection of spurious diffusion peaks and longer windows (e.g., 35, 50 frames) causing averaging of diffusion peaks. Diffusion coefficients (*D*) were calculated for each window using a MATLAB class called msdanalyzer (***Tarantino et al., 2014***), where *D* was estimated from the linear regression fitting of the MSD plots of rolling windows using the first five time points. *D* is calculated as:

$$D = \frac{1}{2d} \times \frac{MSD\,(dt)}{dt}$$

**Table 1.** crRNA sequences for dCas9-binding oligonucleotide sequences used for lambda DNA preparation.

| Identity | Sequence |
| --- | --- |
| Cas9 crRNA sequence 'lambda 1' | 5'-/ AltR 1/rGrUrG rArUrA rArGrU rGrGrA rArUrG rCrCrA rUrGrG rUrUrU rUrArG rArGrC rUrArU rGrCrU / AltR2/-3' |
| Cas9 crRNA sequence 'lambda 2' | 5'-/ AltR 1/rCrUrG rGrUrG rArArC rUrUrC rCrGrA rUrArG rUrGrG rUrUrU rUrArG rArGrC rUrArU rGrCrU / AltR2/-3' |
| Cas9 crRNA sequence 'lambda 3' | 5'-/AltRl /rCrArG rArUrA rUrArG rCrCrU rGrGrU rGrGrU rUrCrG rUrUrU rUrArG rArGrC rUrArU rGrCrU / AltR2/-3' |
| 3x-biotin-cos1 oligo | 5'-/5Phos/ AGG TCG CCG CCC TT/iBiodT/TT/iBiodT/TT/3BiodT/-3' |
| 3x-digoxigenin-cos2 oligo | 5'-/5Phos/ GGG CGG CGA CCT TT/iDigN/TT/iDigN/TT/3DigN/-3' |
| Adapter oligo for lambda DNA dual end biotin labeling | 5'-/5Phos/ GGG CGG CGA CCT TGC A-3' |

where $d$ is the number of dimensions (1 in this case).

Each window was then classified as non-diffusive, low-diffusive, or high-diffusive states. Based on the dCas9 profile as a control for truly immobile particles, we set <0.01 μm²/s as the threshold for the immobile state. The thresholds for low-diffusive ($D < 0.04$ μm²/s) and high-diffusive ($D > 0.04$ μm²/s) states were set at 0.04 μm²/s. The mean of all $D$ values associated with each diffusive state was calculated to obtain $D$ values for the three states per trajectory (*Figure 1C*).

Four crRNAs specific to lambda DNA were used (*Table 1*) to immobilize dCas9 bound to lambda DNA in standard Cas9 cleavage buffer. To quantify state durations and relative percentages of the three diffusive states per trajectory, two additional filtering steps were applied to reduce the detection of spurious diffusive transitions, which mainly arise from tracking errors and noise in raw kymographs. First, the 'smooth()' MATLAB function (default parameter) was used to smooth the position vector using a moving average filtering method. The smoothed data were then used to classify each window into three diffusive states. The 'bwconncomp.m' MATLAB function was used to connect neighboring windows with the same diffusive states into segments of non-diffusive, low-diffusive, and high-diffusive states. A second filtering step removed segments shorter than 10 consecutive windows as spurious detections. Finally, the lengths of each segment were used to compute state durations, and the net lengths of each of the three diffusive segments over the total trajectory length were used to determine the relative percentages of the three diffusive states per trajectory.

## Localization precision calculation

There are various methods to calculate precision, but the most commonly used one is based on calculations of the standard deviation of the Gaussian fit for localization:

$$\sigma_0 \geq \frac{s}{\sqrt{N}}$$

where $s$ is the standard deviation of the Gaussian fit, and $N$ is the total number of photons emitted. According to this relationship, we calculate a precision of greater than 72 nm. This calculation is based on 40 single-molecule trajectories, each approximately 400 data points long.

## Bimolecular remodeler–remodeler imaging and interaction analysis

Remodeler–remodeler interactions were assessed by imaging single-molecule concentrations of remodelers labeled in two colors (red and green) together on the same piece of DNA. The same concentrations of RSC-JFX650 and ISW2-JFX650 were used for the two-color imaging experiments as was used in the single-color imaging experiments.

To define colocalizations, we performed the following steps. Firstly, we calculated the diffusion coefficient for each molecule over time using time windows, specifically utilizing 20 exposure time intervals in our study. Next, we determined the mean diffusion coefficient for each molecule. Based on the mean diffusion coefficients, we determined the maximum displacements that can occur between

two molecules assuming independent movement, employing a simple Brownian motion model. The maximum distance between two molecules was calculated using:

$$x_1 = \sqrt{2\mu_{D1}\Delta t}$$

$$x_2 = \sqrt{2\mu_{D2}\Delta t}$$

where $\mu_1$ and $\mu_2$ represent the mean diffusion coefficients of the first and second molecules, respectively, and $\Delta t$ denotes the camera exposure time. Consequently, the maximum distance between two molecules is given by:

$$d = x_1 + x_2$$

For RSC–RSC interactions, the average threshold was found to be 0.31 μm, and for RSC–ISW2 interactions, it was 0.29 μm.

To reduce noise arising from shot noise, we applied a Gaussian filter to smooth the trajectories, using a time window of five exposure time points. Subsequently, we identified time points at which the distance between molecules, based on the smoothed trajectories, fell below the threshold, enabling the determination of colocalized time points. These colocalized time points were then categorized into short and long colocalizations. Specifically, we set a threshold of five exposure times (the same time window used for trajectory smoothing). If the duration of colocalization exceeded this threshold, it was automatically classified as a long colocalization. For colocalizations below the threshold, we assessed the duration of colocalization and the local diffusion coefficients. If the displacement of the molecular distance exceeded the calculated average displacement, it was categorized as a long colocalization; otherwise, it was considered a short colocalization.

To validate the short colocalizations, we examined the diffusion coefficient for this part of the trajectories, ensuring that the molecular distance remained within the threshold and subsequently exited it. If the calculated diffusion coefficient for this segment was greater than the average diffusion coefficient, we classified the colocalization as a short colocalization; otherwise, it was considered a long colocalization. In summary, our colocalization analysis involved calculating diffusion coefficients, determining thresholds, smoothing trajectories, identifying colocalized time points, categorizing colocalizations into short and long durations, and verifying short colocalizations based on the diffusion coefficient. These steps allowed us to define colocalizations and evaluate their durations in our study (*Figure 2—figure supplement 1A–F*).

Dwell-time analysis was performed using the Akaike information criterion to determine the most probable number of lifetime models. In this case, a single-exponential model was identified. Subsequently, lifetime estimation was carried out using the maximum likelihood estimation (MLE) method with the XYZ package in MATLAB. The likelihood is expressed as

$$P\left(t|\tau\right) = \prod_{i=1}^{I} Exp\left(t_i \ ; \ \tau\right) \ = \prod_{i=1}^{I} \frac{1}{\tau} e^{-\frac{t_i}{\tau}} \ = \ \frac{1}{\tau^I} \exp\left(-\frac{1}{\tau}\sum_{i=1}^{I} t_i\right).$$

## Lambda DNA preparation with biotin and digoxigenin labeling

Lambda DNA was prepared with three biotins on one end and three digoxigenins on the other end using the following protocol. Custom oligos (*Table 1*) were sourced from IDT, and lambda DNA was acquired from NEB (cat# N3011S). Oligo 1 was annealed to lambda DNA by adding a 25-fold molar excess to the DNA in an annealing buffer containing 30 mM HEPES (pH 7.5) and 100 mM KCl. The mixture was heated at 70°C for 10 min and cooled gradually to room temperature. Subsequently, 2 μl of NEB T4 DNA ligase (400 U, cat# M0202S) and its corresponding buffer with ATP were added, followed by a 30-min incubation at room temperature. A 50-fold molar excess of oligo 2 was then introduced, along with an additional 1 μl of T4 DNA ligase and its buffer containing ATP, with volume adjustments. The mixture was incubated at room temperature for another 30 min and heat-inactivated at 65°C for 10 min. The end-labeled lambda DNA was purified using the Qiaex II gel-extraction DNA clean-up kit, as per the manufacturer's instructions (QIAGEN, cat# 20021).

## Lambda DNA preparation with biotin labeling on both ends

Lambda DNA was also prepared with three biotins on each end using an adapter oligo (*Table 1*) to recycle the 3x-Biotin Cos1 Oligo. The protocol mentioned above was followed, with some modifications. An equimolar mixture of adapter oligo and the 3x-Biotin Cos1 Oligo were annealed by heating and gradually cooling using IDT protocols and buffers. This adapted oligo was then used instead of oligo 2 in the aforementioned protocol.

## Lambda nucleosome array preparation

A salt gradient dialysis method was employed to reconstitute nucleosomes onto lambda DNA, using optimized laboratory procedures based on established protocols (*Luger et al., 1999*; *Vary et al., 2003*). The buffers used in this reconstitution included high-salt buffer (10 mM Tris–HCl pH 7.5, 1 mM EDTA pH 8.0, 2 M NaCl, 5 mM 2-mercaptoethanol [BME]) and low-salt buffer (10 mM Tris–HCl pH 7.5, 1 mM EDTA pH 8.0, 50 mM NaCl, 5 mM BME). Cy3-labeled H2A-containing octamer, prepared as formerly described (*Ranjan et al., 2013*), was titrated onto lambda DNA (either 0.5 or 1 µg DNA) at molar ratios of 40:1, 20:1, 10:1, 5:1, and 2.5:1. Reconstitution reactions were carried out in 10 mM Tris pH 7.5, 1 mM EDTA pH 8.0, 0.1 mg/ml BSA Roche (cat# 10711454001), and 1 mM BME. A 16-hr dialysis was set up using a 7-kDa MWCO Slide-A-Lyzer MINI Dialysis Device (cat# 69560), placed in a flotation device in high-salt buffer. Low-salt buffer was gradually added to the high-salt buffer throughout the dialysis, with constant stirring. After the dialysis period, the solution was replaced with 100% low-salt buffer and allowed to dialyze for an additional hour. The reconstitution efficiency was assessed using an electrophoretic mobility shift assay by running lambda nucleosome arrays on a 0.5% agarose gel in 0.5× Tris-borate-EDTA (TBE) (*Figure 3—figure supplement 1A*).

## Histone labeling, octamer reconstitution, and purification

H2A(K120C)/H2B dimers, prepared in-house, were labeled with maleimide-JFX554 using a standard protein maleimide labeling protocol with a 50-fold molar excess of dye to protein. The reaction was allowed to proceed for 3 hr at room temperature before quenching with BME. A small aliquot was reserved to assess labeling efficiency, and excess free dye was removed through three successive buffer exchanges using a 10 K MWCO Amicon spin filter. Labeling efficiency was estimated at 61% based on nanodrop spectral absorption readings at 554 and 280 nm, and the extinction coefficients of the dye and protein. The labeled H2A/H2B dimer was then denatured overnight in denaturing buffer (7 M guanidinium chloride, 10 mM DTT, 20 mM Tris pH 7.5). The following day, a 1.5-fold molar excess of denatured H2A/H2B was mixed with H3 and H4, both solubilized in denaturing buffer, and incubated for 1 hr at room temperature. 0.5 mg of histones H2A and H2B were used in this step.

Octamer refolding was conducted by successively dialyzing the mixture into 1 l of refolding buffer (2 M NaCl, 5 mM BME, 1 mM EDTA, 10 mM Tris–HCl pH 7.5) four times, with each dialysis separated by approximately 12 hr. The samples were then fractionated by size-exclusion chromatography on a Superdex 200 column equilibrated with refolding buffer using an AKTA Fast Protein Liquid Chromatograph (FPLC). Octamer fractions were collected, and SDS–PAGE was employed to determine which peak fractions to pool. A second estimation of labeling efficiency at 68% was obtained after octamer purification. Octamer samples were stored at 1 mg/ml concentrations in 2 M NaCl and 50% glycerol at −20°C until further use.

## Lambda nucleosome array imaging and validation

Lambda nucleosome arrays (biotin labeled on both ends) were captured by oscillating the distance between two trap centers (traps 1 and 2 with a ~4.38 µm diameter streptavidin-coated polystyrene bead under buffer flow (<0.2 bar)) containing approximately 63 pg/µl lambda nucleosome array. When the force measured on trap 2 exceeded 5 pN, the oscillation and buffer flow were stopped. The distance between the beads was adjusted to achieve a force of ~0 pN on trap 2, and the traps were moved into the protein channel. Buffer flow was maintained at <0.1 bar for a maximum of 30 s to refresh protein in the channel and promote remodeler binding. This flow did not disrupt the nucleosome array stability. The DNA was then pulled to a tension of 5pN, at which point the distance between the two traps was fixed and imaging was started. For imaging Cy3-labeled nucleosomes, a time-lapse scheme in the green channel was used to preserve fluorescence signal lifetime, despite poorer photostability compared to JFX-labeled remodelers. Green excitation (15% of maximum) was

intermittently applied to visualize nucleosome positions on the array, with approximately 1 s of green excitation used at various time points. For imaging JFX554-labeled nucleosomes, initially the same intermittent excitation protocol was implemented, but was soon after switched to continuous excitation at lower laser intensity (10% of maximum) due to the brightness and superior photostability of the JFX dye. To note, JFX554-labeled nucleosomes blink as well as dim in and out their intensity prior to an ultimate photobleaching event (visible in *Figure 4D*). For imaging JFX650-labeled remodelers, red excitation (6.5% of maximum) was applied continuously. After JFX650-labeled remodelers bleached, green excitation continued until the remaining green signal photobleached. Lambda nucleosome arrays contained a variable number of nucleosomes. To determine the mean number of nucleosomes per array, the following procedure was employed. After imaging, the lambda nucleosome array was moved to a buffer-only channel, and nucleosomes were forcibly unwrapped. A force-clamp between 15 and 20 pN was applied to visualize individual unwrapping events (*Figure 3B, C*). Once unwrapping events ceased, the force was increased to 40 pN and then 60 pN + to denature the DNA. If the DNA remained intact, a second force–distance curve was collected to visualize DNA free of nucleosomes.

## Remodeler–nucleosome colocalization analysis pipeline

To colocalize remodeler signals with nucleosome signals, we first processed nucleosome signals by performing linking analysis and marking signals to be excluded from colocalization based on specific criteria. As mentioned previously, the green laser was pulsed to extend the fluorescence lifetime of Cy3-labeled nucleosomes, which constituted 2/3 of the data collected in this study. A JFX-nucleosome was created to extend the fluorescence lifetime of the nucleosome, achieving a duration comparable to that of the remodeler. For data involving pulsed green lasers, we extended nucleosome positional information by using the last visible fluorescence signal before laser pulsing or photobleaching. This approach enabled generation of positional information for all nucleosome datasets, regardless of fluorophore used, facilitating remodeler–nucleosome colocalization analysis.

Nucleosome fluorescent signals were omitted from consideration if unstable, likely due to non-specific adsorption of fluorescently labeled histones remaining after reconstitution of lambda nucleosome arrays. These signals were also considered in linking analysis, and stable signals that later linked to an unstable signal were removed from colocalization analysis. Remodelers colocalizing within 0.17 pixels (~500 bps) of the nucleosome signal were considered colocalized, and their molecular identifiers were recorded for immobility analysis.

## Translocation analysis pipeline

Kymographs obtained in the presence of 1 mM ATP were evaluated for translocation based on directional motion exhibited by either remodeler or nucleosome signals, or their colocalization. Traces were manually categorized into segments displaying 1D search, stable non-translocating engagement, or translocation. Stable nucleosome colocalized signals in kymographs with 1 mM ATPγS were used as a control condition, as translocation should not occur due to its ATP hydrolysis dependence. Time trace information for translocating segments was fitted to a linear regression model. In cases of poor fits, traces were re-evaluated for speed changes. Translocation events with speed changes were fitted separately using linear regressions for each segment of uniform speed. Translocation events lasting less than 5 s, spanning less than 300 bps, or exhibiting an $R^2$ value below 0.5 were excluded from reporting, as they could not be distinguished from fits in the ATPγS control condition.

## Acknowledgements

We thank Raquel Merino Urteaga for kindly sharing her 80N3[Cy5-DNA; Cy3-H2A] nucleosome, Anand Ranjan for experimental guidance with octamer preparation and nucleosome reconstitution, Robert K Louder for consultation on the structural basis of nucleosome pushing and pulling and our laboratory colleagues for comments. This study was supported by funds from the Korea Foundation for Advanced Studies Fellowship (JMK), the National Science Foundation DGE-1746891 (CC), the National Institute of Health grants GM132290 (CW), GM149291 (CW), GM122569 (TH), T32 GM007445 (CC), S10 OD025221 (TH), the Johns Hopkins Bloomberg Distinguished Professorship (TH, CW), and Howard Hughes Medical Institute (TH).

## Additional information

### Competing interests

Jonathan B Grimm: US Patent 11,091,643 describing deuterated fluorophores and variant compositions are assigned to HHMI. Luke D Lavis: Founder and shareholder of Eikon Therapeutics. US Patent 11,091,643 describing deuterated fluorophores and variant compositions are assigned to HHMI. The other authors declare that no competing interests exist.

### Funding

| Funder | Grant reference number | Author |
| --- | --- | --- |
| Korea Foundation for Advanced Studies | Fellowship | Jee Min Kim |
| National Science Foundation | DGE-1746891 | Claudia C Carcamo |
| National Institutes of Health | GM132290 | Carl Wu |
| National Institutes of Health | GM149291 | Carl Wu |
| National Institutes of Health | GM122569 | Taekjip Ha |
| National Institutes of Health | T32 GM007445 | Claudia C Carcamo |
| National Institutes of Health | S10 OD025221 | Taekjip Ha |
| Bloomberg Distinguished Professorship | | Taekjip Ha Carl Wu |
| Howard Hughes Medical Institute | | Taekjip Ha |

The funders had no role in study design, data collection, and interpretation, or the decision to submit the work for publication.

### Author contributions

Jee Min Kim, Claudia C Carcamo, Conceptualization, Resources, Data curation, Software, Formal analysis, Investigation, Visualization, Methodology, Writing - original draft, Writing – review and editing; Sina Jazani, Software, Formal analysis; Zepei Xie, Resources, Investigation; Xinyu A Feng, Matthew Poyton, Katie L Holland, Jonathan B Grimm, Luke D Lavis, Resources; Maryam Yamadi, Validation, Investigation; Taekjip Ha, Supervision, Funding acquisition, Project administration, Writing – review and editing; Carl Wu, Conceptualization, Supervision, Funding acquisition, Project administration, Writing – review and editing

### Author ORCIDs

Jee Min Kim http://orcid.org/0000-0002-9353-6431
Claudia C Carcamo http://orcid.org/0000-0002-2646-188X
Xinyu A Feng http://orcid.org/0000-0001-5862-1983
Maryam Yamadi http://orcid.org/0000-0001-7080-5566
Matthew Poyton https://orcid.org/0000-0003-1261-2138
Katie L Holland http://orcid.org/0009-0007-1183-195X
Luke D Lavis https://orcid.org/0000-0002-0789-6343
Taekjip Ha http://orcid.org/0000-0003-2195-6258
Carl Wu http://orcid.org/0000-0001-6933-5763

Review #1 (Public Review) https://doi.org/10.7554/eLife.91433.3.sa1
Review #2 (Public Review) https://doi.org/10.7554/eLife.91433.3.sa2
Author Response https://doi.org/10.7554/eLife.91433.3.sa3

## Additional files

### Supplementary files
• MDAR checklist

### Data availability

We have made raw data, source code, and corresponding tables associated with this study available through Dryad, an online data repository [Kim, Jee Min et al. (2024), Dynamic 1D Search and Processive Nucleosome Translocations by RSC and ISW2 Chromatin Remodelers, Dryad, Dataset, https://doi.org/10.5061/dryad.jsxksn0g5]. The uploaded raw data consist of tiff images of kymographs along with associated csv files of single-particle tracking for both color channels. The source code provided is written in MATLAB and be used with the csv files to calculate diffusion coefficients using the rolling-window approach described in the methods. We have also included a csv file containing tables that correspond to the data presented in the figures; these tables include diffusion coefficient values as well as the respective *n*-values where relevant.

The following dataset was generated:

| Author(s) | Year | Dataset title | Dataset URL | Database and Identifier |
|---|---|---|---|---|
| Kim JM, Carcamo C, Jazani S, Xie Z, Feng X, Yamadi M, Poyton M, Holland K, Grimm J, Lavis L, Ha T, Wu C | 2024 | Dynamic 1D Search and Processive Nucleosome Translocations by RSC and ISW2 Chromatin Remodelers | https://doi.org/10.5061/dryad.jsxksn0g5 | Dryad Digital Repository, 10.5061/dryad.jsxksn0g5 |

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
