## [Editor Report · eLife assessment]

This manuscript describes **fundamental** single-molecule correlative force and fluorescence microscopy experiments to visualize the 1D diffusion dynamics and long-range nucleosome sliding activity of the yeast chromatin remodelers, RSC and ISW2. **Compelling** evidence shows that both remodelers exhibit 1D diffusion on bare DNA but utilize different mechanisms, with RSC primarily hopping and ISW2 mainly sliding on DNA. These results will be of interest to researchers working on chromatin remodeling.

---

## [Referee Report · Review #1 (Public Review)]

Single-molecule visualization of chromatin remodelers on long chromatin templates-a long sought-after goal-is still in its infancy. This work describes the behaviors of two remodelers RSC and ISW2, from SWI/SNF and ISWI families respectively, with well-conducted experiments and rigorous quantitative analysis, thus representing a significant advance in the field of chromatin biology and biophysics.

---

## [Referee Report · Review #2 (Public Review)]

The authors use a dual optical trap instrument combined with 2-color fluorescence imaging to analyze the diffusion of RSC and ISW2 on DNA, both in the presence and absence of nucleosomes, as well as long-range nucleosome sliding by these remodelers. This allowed them to demonstrate that both enzymes can participate in 1D diffusion along DNA for rather long ranges, with ISW2 predominantly tracking the DNA strand, while RSC diffusion involves hopping. In an elegant two-color assay, the authors were able to analyze interactions of diffusing remodeler molecules, both of the same or different types, observing their collisions, co-diffusion and bypassing. The authors demonstrate that nucleosomes act as barriers for remodeler diffusion, either repelling or sequestering them upon collision. In the presence of ATP, they observed surprisingly processive unidirectional nucleosome sliding with a strong bias in the direction opposite to where the remodeler approached the nucleosome from for ISW2. These results have fundamentally important implications for the mechanism of nucleosome positioning at promoters in vivo, will be of great interest for the scientific community, and will undoubtedly spark exciting future research

---

## [Author Response]

The following is the authors’ response to the original reviews.

**eLife assessment**
This manuscript describes fundamental single-molecule correlative force and fluorescence microscopy experiments to visualize the 1D diffusion dynamics and long-range nucleosome sliding activity of the yeast chromatin remodelers, RSC and ISW2. Compelling evidence shows that both remodelers exhibit 1D diffusion on bare DNA but utilize different mechanisms, with RSC primarily hopping and ISW2 mainly sliding on DNA. These results will be of interest to researchers working on chromatin remodeling.
**Reviewer #1 (Public Review):**
Single-molecule visualization of chromatin remodelers on long chromatin templates-a long sought-after goal-is still in its infancy. This work describes the behaviors of two remodelers RSC and ISW2, from SWI/SNF and ISWI families respectively, with well-conducted experiments and rigorous quantitative analysis, thus representing a significant advance in the field of chromatin biology and biophysics. Overall, the conclusions are supported by the data and the manuscript is clearly written. However, there are a few occasions where the strength of the conclusion suffers from low statistics. Some of the statements are too strong given the evidence presented.

We thank the reviewer for the thorough and considerate review of our manuscript. We have increased the statistics when possible and have toned down the conclusions wherever further experimentation to improve statistics could not be done expeditiously.

Specific Comments:(1) It is confusing what is the difference between the "non-diffusive" behavior of the remodeler upon nucleosome encounter and the nucleosome-translocating behavior in the presence of ATP. For example, in Figure 3F, readers can see a bit of nucleosome translocation in the first segment. Is the lower half-life of "non-diffusive" ISW2 with ATP on a nucleosome array because it is spending more time translocating nucleosomes? The solid and dashed green lines in Figure 3F and 3G are not explained. It is also not explained why Figure 3H and 3I are fit by double exponentials.

We thank the reviewer for calling upon us to clarify these points. In both the case of translocation and stable non-translocating colocalization, the chromatin remodeler is marked as “non-diffusive” because the molecule is not moving quickly enough to be detected by our rolling-window (20 frames considered) diffusion coefficient analysis. We have updated the text to point out the translocation that is occurring in the panels indicated and noted that this type of motion is not detected by our automated analysis. Thus, translocation events were manually segmented for analysis from kymographs; a note of this was added to the results section (Results section # 1; Paragraph # 2).

To address the question of whether the half-life of “non-diffusive” ISW2 with ATP on the nucleosome array is because of increased time spent in translocation, we have computed the percentage of “non-diffusive” time spent translocating in the presence of ATP for both remodelers; for ISW2, 14% of “non-diffusive” times are translocation whereas for RSC, 28% of “non-diffusive” times are translocation. Given that these percentages are not negligible, the reviewer helped identify an important parameter that better describes the effects of ATP hydrolysis on nucleosome binding for ISW2. In addition, we computed and compared the half-life of translocation times for both remodelers to the “non-diffusive” times and found that RSC translocates with a half-life of 20 s (similar to the half-life of “non-diffusion”) whereas ISW2 translocates with a half-life of 17 s (longer than the half-life of “non-diffusion”). We believe that this new information improves understanding of the role of ATP hydrolysis in turning over ISW2-nucleosome binding interactions, which result in the shorter “non-diffusive” lifetime as well as the shorter and more rarely observed ISW2 translocation events. We have updated the text to include these observations and our interpretation (Results section # 3; Paragraph # 3). As was already included in the text (Results section # 3; Final Paragraph), we speculate that this behavior may be due to a hydrolysis-dependent turnover of the ISW2-nucleosome bound state and refer the reader to Tim Richmond’s 2004 EMBO paper titled “Reaction cycle of the yeast Isw2 chromatin remodeling complex” in which bulk experiments show that ATP hydrolysis affects ISW2-nucleosome bound lifetimes.

We thank the reviewer for also pointing out where details were missing from the figure legend and results section regarding Figure 3. We have added a description of the dashed and solid lines to the figure legend (Figure 3; Legend). We have also described why Figures 3H and I are fit to double exponentials to the results section (Results section # 3; Paragraph # 2).

(2) What is the fraction of 1D vs. 3D nucleosome encountered by the remodelers? This is an important parameter to compare between RSC and ISW2.

We thank the reviewer for raising this point. We agree that this is an important parameter to compare between RSC and ISW2; knowledge of this parameter would enable quantitative predictions to be made from our data regarding target localization efficiency increases owed to 1D scanning for each remodeler. We regretfully could not quantify this due to technical limitations of our measurements. A note about this limitation along with an explanation for why we were unable to quantify this parameter have been added to the main text (Results section # 3; end of Paragraph # 1).

(3) A major conclusion stated repeatedly in the manuscript is that nucleosome translocation by a remodeler is terminated by a downstream nucleosome. But this is based on a total of 4 events. The problem of dye photobleaching was mentioned, which is a bit surprising considering that the green excitation was already pulsed. The authors should try to get more events by lowering the laser power or toning down the conclusion that translocation termination is prominently due to blockage by a downstream nucleosome. Quantifying the translocation distances before termination, in addition to the durations (Figure 4G and 4H), would also be helpful.

We thank the reviewer for these observations and feedback. We agree that only 4 observations of direct visualization of remodeler translocation termination by a downstream nucleosome is a small n-value, and have chosen to omit presentation of these rare events in the manuscript.

(4) The claim on nucleosome translocation directionality is also based on a small number of events, particularly for RSC. 6/9 is hardly over 50% if one considers the Poisson counting error (RSC was also found to switch directions.) If the authors would like to make a firm statement to support the "push-pull" model, they should obtain more events.

We thank the reviewer for this critique and agree with the reviewer’s concern. In addition to adding data from two additional experimental replicates of RSC nucleosome translocation (which had the smaller n-value), we have also re-evaluated all events containing translocation for additional evidence in support or against the “push-pull” model. Previously we were only considering events where 1D diffusion on DNA leads immediately to translocation. Now we add the following categories to the count: (1) events where translocation terminates with the remodeler dissociating from the nucleosome and performing a 1D diffusive search, (2) events where 1D diffusion on DNA leads to association with a nucleosome and after a paused colocalization we observe translocation, and (3) the inverse scenario of (2) (see schematics in Figure 5 – figure supplement 1). These new results, detailed below, are now included in place of the older results in (Results Section # 5; Paragraph # 2). Furthermore, we toned down our argument and clarified that a larger n-value would be needed to be definitive, especially since we observe RSC switching directions, as the reviewer points out.

By aggregating in new RSC data and using only events where 1D diffusion leads immediately to translocation, we observe 10/12 events in support of the “push” model. If we include these other categories in addition to aggregating the previous data with the new data, a total of 20/25 events are in support of the “push” model. For RSC, the breakdown in the other categories was as follows: (1) 7/10 events, (2) 1/1 events with a paused time of 5 seconds, and (3) 2/2 events with a paused time of 36 and 50 seconds.

For ISW2, we had previously reported 12/13 events where 1D search lead immediately to translocation. After combing through the data a second time, we decided to omit two events which were less clear; Now we report 10/11 events in support of the “pull” model from this initial category. If we include these other categories in addition to the original, a total of 19/21 events are in support of the “pull” model. For ISW2, the breakdown in the other categories was as follows: (1) 4/4 events, (2) 4/4 events with pause times of 44, 27, 29, and 8 seconds, (3) 1/2 events with paused times of 5 and 19 seconds.

(5) At 5 pN of tether tension, the outer wrap of nucleosomes is destabilized, which could impact nucleosome translocation dynamics. Additionally, a low buffer flow was kept on during data acquisition, which could bias remodeler diffusion behavior. The authors should rule out or at a minimum discuss these possibilities.

We thank the reviewer for raising the important point regarding outer wrap destabilization of the nucleosome occurring at 5pN of tension. We have added an additional section to the discussion that reviews the literature on tension effects on nucleosome stability as well as what is currently known of the effects of tension on remodeler translocation on DNA (Discussion Paragraph # 3). While we cannot exclude the possibility that the 5pN of tension used in this study is a causative factor of the observed fast speed or high processivity nucleosome translocation that we report, we believe that with the modifications made to the text to emphasize to the reader of these possibilities, the reader can draw informed conclusions on the significance of our findings. The topic of force effects on remodeling outcomes is an interesting subject for the future.

We apologize that the experimental details on buffer flow used during imaging was unclear in our initial submission; we do not have buffer flowing during imaging, rather the buffer containing protein is flowed over the DNA at low pressure just prior to imaging. The flow is completely stopped before the DNA or nucleosome array is stretched to 5pN of tension for imaging (See Methods section: Single Molecule Tracking and Analysis).

**Reviewer 1 (Recommendations For The Authors):**
(1) The figure panels could be better arranged to focus on the main messages of the paper.(i) Figure 3C-E should go to a supplemental figure.

We thank the reviewer for this helpful suggestion. As recommended, we moved Figure 3C to the supplemental figure as this panel did not pertain to the main message of the paper.

(ii) Figure 4 could be split into two figures, one characterizing processive nucleosome translocation (4C, D, G, H, I, J, K, and relevant panels in S4), and the other showing the differential directionality of each remodeler (4E, F, L, and relevant panels in S4).

We thank the reviewer for their suggestions that help better organize our presentation of the data. As the reviewer suggests, we split figure 4 into two figures: figure 4 which now focuses on translocation characterization and figure 5 which now focuses on the differential directionality of each remodeler.

(iii) The nucleotide condition should be clearly indicated in the figures or legends. For example, it is unclear if the data in Figure 2 were generated with or without ATP.

We thank the reviewer for taking note of this. We have added clear indications of the nucleotide condition to figures where this is relevant, including in Figure 2 as indicated.

(iv) There are many cartoon panels, and some are redundant (e.g., Figure 1A and 1B, Figure 3A and 3B).

We thank the reviewer for bringing up this point. We agree that some cartoons are redundant. We have eliminated Figure panel 1B and Figure panel 3A of the original figures from the new figures.

(2) The last paragraph of the Results section should be moved to Discussion. This paper did not directly address the effects of RSC/ISW2 on NDR length.

We thank the reviewer for this suggestion. We agree and have moved the last paragraph of the Results section to the Discussion..

(3) There are some typos in the text. For example, "Of the two main types of 1D diffusion, hopping and sliding" is not a complete sentence.

We thank the reviewer for catching this typo and bringing our attention to others. Upon a more careful proofreading of the text and figures we have caught and amended this and other typos.

(4) What are the green lines in Figure S1F?

We thank the reviewer for asking this question. The green lines were meant emphasize how the percentage of traces in the majority high diffusion category increases for RSC but not for ISW2 in response to increases in the KCl concentration. Since this was confusing, we removed these green lines.

**Reviewer # 2 (Public Review):**
Summary:The authors use a dual optical trap instrument combined with 2-color fluorescence imaging to analyze the diffusion of RSC and ISW2 on DNA, both in the presence and absence of nucleosomes, as well as long-range nucleosome sliding by these remodelers. This allowed them to demonstrate that both enzymes can participate in 1D diffusion along DNA for rather long ranges, with ISW2 predominantly tracking the DNA strand, while RSC diffusion involves hopping. In an elegant two-color assay, the authors were able to analyze interactions of diffusing remodeler molecules, both of the same or different types, observing their collisions, co-diffusion, and bypassing. The authors demonstrate that nucleosomes act as barriers for remodeler diffusion, either repelling or sequestering them upon collision. In the presence of ATP, they observed surprisingly processive unidirectional nucleosome sliding with a strong bias in the direction opposite to where the remodeler approached the nucleosome from for ISW2. These results have fundamentally important implications for the mechanism of nucleosome positioning at promoters in vivo, will be of great interest to the scientific community, and will undoubtedly spark exciting future research.Strengths:The mechanism of target search for chromatin-interacting protein machines is a 'hot' topic, and this manuscript provides extremely important and timely new information about how RSC and ISW2 find the nucleosomes they slide. Intriguingly, although both remodelers analyzed in this study can diffuse along DNA, the diffusion mechanisms are substantially different, with extremely interesting mechanistic implications.The strong directional preference in nucleosome sliding by ISW2 dictated by the direction it approaches the nucleosomes from during 1D sliding on DNA is a very intriguing result with interesting implications for the regulation of nucleosome organization around promoters. It will be of great interest to the scientific community and will undoubtedly inspire future research.Relatively little is known about nucleosome sliding at longer ranges (>100bp), and this manuscript provides a unique view into such sliding and also establishes a versatile methodology for future studies.Weaknesses:All measurements were conducted at 5pN tension, which induces unwrapping of the outer DNA gyre from nucleosomes. This could potentially represent a limitation for experiments involving nucleosomes, since partial nucleosome unwrapping could affect the behavior of remodelers, especially their sliding of nucleosomes.

We thank the reviewer for succinctly summarizing the strengths and weaknesses of our study. We have changed the Discussion to better review the literature on the effects of 5pN of tension on nucleosome wrapping and have more clearly presented the limitations of our studying owing to our conducting measurements at 5pN of tension. In doing so, we have tried to emphasize the strengths of our study identified by the reviewer and better inform the reader of the weaknesses.

**Reviewer #2 (Recommendations For The Authors):**
Although not required, nucleosome sliding data under lower tensions (e.g., < = 2pN) could be a valuable addition to the manuscript. Indeed, to my knowledge, there is no data on force-dependent rates of nucleosome sliding, so a conclusive demonstration of changes in remodeling rate with tension would be an exciting new result and might be discussed in the context of a potential tension in chromatin. If such experiments cannot readily be added, the authors could alternatively discuss this potential limitation in more detail.

We thank the reviewer for this suggestion. We agree that adding data at lower tensions (< = 2pN) would have been valuable. Due to time constraints, this will be the subject for the future. We agree that knowledge of the effects of tension would be especially interesting in light of the possibility that tension on chromatin in cells may be affecting remodeler function. We have added a discussion of this potential significance of future work to the discussion (Discussion Section; Paragraph # 3). We have also elaborated on the potential limitation of only conducting measurements at 5pN to the discussion (Discussion Section; Paragraph # 3), as the reviewer recommends.

The quantitative implications of the proposed mechanism for targeting ISW2 and RSC towards +1 and -1 nucleosomes are highly interesting. To further strengthen the mechanistic implications, the authors could consider quantitatively analyzing how the observed 1D diffusion would affect the probabilities of binding to +1 and -1 versus to other nucleosomes.

We thank the reviewer for their thoughtful suggestion. While we would have liked to present a final quantitative model that integrates the experimental parameters on 1D diffusion that we present in this study with the parameters extracted from live cell single particle tracking studies, there are key parameters for model building that are missing from our study, due to technical limitations. Namely, we were not able to quantify the fraction of 1D vs 3D nucleosome encounters by remodelers, because the majority of the protein that we image has been bound before the start of imaging; very few proteins bind the nucleosome arrays after the start of imaging as the protein concentration in the imaging chamber is very low. This makes observing binding directly to a nucleosome a very rare event, especially due to the sparse density of nucleosomes (~10) on the array (~50,000 kb).

The low-diffusion state is intriguing - could the authors speculate about the nature of this state?

We thank the reviewer for the question. We had added some speculation about the nature of the low-diffusion state to the results section (Results Section # 1; Paragraph 2). One thought that we have is that this may be due to more stable interactions made between remodelers and free DNA when they become trapped in a conformation state that binds more tightly to DNA. Conformational changes may result in different scanning speeds for chromatin remodelers; e.g. SWR1 was shown to scan DNA quicker when bound to ATP (Carcamo, C. et al. eLife 2022). Another possibility is that certain sequences due to their intrinsic curvature, for instance, or their AT-content may trap the remodeler which may make more contacts with the DNA at these sites.

Minor points:Information on the labeling efficiencies for the remodelers would be helpful.

We thank the reviewer for pointing this out. We assessed labeling saturation by running gels of remodeler labeling with increasing molar ratios of dye to protein and did not observe increased labeling efficiency above the molar ratio used for proteins imaged in our study (see added Figure 1 – figure supplement 1, panel A). From this, we assessed that we have high protein labeling efficiency. We could not assess the labeling efficiency using the standard absorbance method as the extinction coefficient for JFX650 was measured with 1% v/v TFA (PMCID: PMC8154212) which is not compatible for use in assessing our protein labeling efficiency in an aqueous buffer.

How were the experimental conditions adjusted for two-color diffusion experiments in order to optimize the probability of observing two remodeler molecules with different labels at the same time.

We thank the reviewer for this clarifying question. To image both remodelers on the same DNA, we combined the remodelers using the same concentrations that produced single molecule densities when the remodelers were imaged separately. We have clarified this point in the Methods section: “Bimolecular Remodeler-Remodeler Imaging and Interaction Analysis”.

The authors should check the figures for consistency of labeling and provide definitions for abbreviations used in them (e.g. CDF and PDF).

We thank the reviewer for catching inconsistencies in labeling in our figures. We have updated the figures such that there is consistent labeling throughout. We have also provided definitions for abbreviations such as Cumulative Distribution Function (CDF) and Probability Distribution Function (PDF) in the figure legends where applicable.

In the section "Remodeler-remodeler collisions during 1D search" (4th line from the end) reference to Fig3D seems to be out of place.

We thank the reviewer for catching this typo. We have reworded this section such that each figure panel can be discussed sequentially, eliminating this out of place reference to Fig 3D.